# URVA and Local Mode Analysis of an Iridium Pincer Complex Efficiently Catalyzing the Hydrogenation of Carbon Dioxide

**Marek Freindorf** [†] **and Elfi Kraka** *,[†]

Chemistry Department, Southern Methodist University, 3215 Daniel Avenue, Dallas, TX 75275-0314, USA
* Correspondence: ekraka@smu.edu
† These authors contributed equally to this work.

**Abstract:** The catalytic effects of iridium pincer complexes for the hydrogenation of carbon dioxide were investigated with the Unified Reaction Valley Approach (URVA), exploring the reaction mechanism along the reaction path traced out by the reacting species on the potential energy surface. Further details were obtained with the Local Mode Analysis performed at all stationary points, complemented by the Natural Bond Orbital and Bader's Quantum Atoms in Molecules analyses. Each of the five reaction paths forming the catalytic cycle were calculated at the DFT level complemented with DLPNO-CCSD(T) single point calculations at the stationary points. For comparison, the non-catalytic reaction was also investigated. URVA curvature profiles identified all important chemical events taking place in the non-catalyzed reaction and in the five reactions forming the catalytic cycle, and their contribution to the activation energy was disclosed. The non-catalytic reaction has a large unfavorable activation energy of 76.3 kcal/mol, predominately caused by HH bond cleave in the $H_2$ reactant. As shown by our study, the main function of the iridium pincer catalyst is to split up the one–step non-catalytic reaction into an energy efficient multistep cycle, where HH bond cleavage is replaced by the cleavage of a weaker IrH bond with a small contribution to the activation energy. The dissociation of the final product from the catalyst requires the cleavage of an IrO bond, which is also weak, and contributes only to a minor extent to the activation energy. This, in summary, leads to the substantial lowering of the overall activation barrier by about 50 kcal/mol for the catalyzed reaction. We hope that this study inspires the community to add URVA to their repertoire for the investigation of catalysis reactions.

**Keywords:** unified reaction valley approach; local mode analysis; iridium-pincer complexes; catalysis; hydrogenation of carbon dioxide



## 1. Introduction

One of the key causes of global warming is our excessive production of $CO_2$ [1–4]. Therefore, a lot of efforts have been put into the exploration of the use of this free carbon source for the commercial synthesis of chemicals and for fuel production [5–11], as well as the transformation of $CO_2$ into functional organic molecules [12–23]. Formic acid is one of the target molecules obtained via $CO_2$ hydrogenation, because of its importance in organic synthesis and its applications in hydrogen storage. Recent research has shown that formic acid can serve as a $H_2$ storage reservoir via its decomposition into $CO_2$ and $H_2$ and the reverse transformation [24–27]. A substantial number of investigations on $CO_2$ hydrogenation in homogenous catalysis have been reported over the last two decades [28–42].

The Nozaki group identified six-coordinate Ir(III)–PNP trihydride (PNP = 2,6–bis(di–isopropylphosphinomethyl)pyridine) as a highly active catalyst with high turn over frequencies (TOF) and turnover number (TON), reflecting the stability of the active site for the $CO_2$ hydrogenation [41,43,44]. The original mechanism of the $CO_2$ hydrogenation to formic acid was reported to include three major steps, (i) $CO_2$ addition to the catalyst, (ii) formic acid dissociation, and (iii) $H_2$ addition in order to regenerate the catalyst [43]. In

the following years, the mechanism was extended including, e.g., basic conditions leading to a more complex catalytic cycle and a larger number of intermediates confirmed by DFT calculations [44–46]. Because the insertion of $CO_2$ into the metal hydride is the crucial step of the catalysis [47,48], possible $CO_2$ insertion into a five-coordinate Iridium(III) dihydride complex as alternative was experimentally investigated [49], showing the formation of $\kappa^2$–bound formate monohydride products by selective electro-catalytic reduction of $CO_2$. The mechanism of the $CO_2$ insertion into five-coordinate Iridium(III) dihydrides and four-coordinate Iridium(I) hydrides was also analyzed theoretically [50], exploring the different reactivity of these complexes with different iridium oxidation states. The catalytic reduction of $CO_2$ to methane with silanes using Brookhart's cationic Ir(III) pincer complex was investigated theoretically [51], focusing on the transfer of silane hydrogens to $CO_2$ and the formation of silylformate, bis(silyl)acetal, methoxysilane and methane. The importance of a hydrogen bond donor in the secondary coordination sphere of the Ir(III)–PNP catalyst was analyzed theoretically [52], finding that the isolation of the formate product from the reaction is straight forward and therefore makes this reaction favorable. $CO_2$ hydrogenation was also investigated using a series of modifications of the Iridium complexes bearing sophisticated N^N–bidentate ligands [53] showing importance of hydroxy groups as proton-responsive substituents in $CO_2$ hydrogenation reactions. Other modifications of the iridium complexes included amide–based ligands, showing that the electron-donating effects of an anionic nitrogen atom and the presence of an OH group near the metal center improves the catalytic activity for $CO_2$ hydrogenation [54]. Although these studies provide valuable insights, a clear understanding of the mechanistic details of each reaction step of the $CO_2$ hydrogenation catalytic cycle and how the catalyst actually lowers the activation barrier has been missing so far.

Therefore, we investigated in this study all major chemical events taking place during the $CO_2$ hydrogenation to formic acid, for the original catalytic cycle proposed by Nozaki [43], using as computational tools the Unified Reaction Valley Approach (URVA) and the Local Modes Analysis (LMA) developed in our group, which are both based on vibrational spectroscopy. URVA requires the accurate and precise determination of the reaction path starting at the transition state (TS) and moving far down into both the entrance and exit channels, a time consuming enterprise. Therefore, we simplified the original Ir(III)-PNP complex in our study by replacing the isopropyl groups with hydrogen atoms. The catalytic cycle involves the five reaction steps **R1**–**R5** shown in Figure 1. For comparison we also included the non-catalyzed reaction **R0**. Reaction movies illustrating the movements of all atoms of the complexes along the reaction path together with the Cartesian coordinates of the stationary points for each of the five reaction paths **R1**–**R5** of the catalytic cycle and the non-catalyzed reaction **R0**, both investigated in the gas phase, are presented in the Supplementary Materials.

**Figure 1.** (**a**) Sketch of the Ir(III)–PNP catalyst; (**b**) non-catalyzed $CO_2$ hydrogenation to formic acid (**R0**); (**c**) Catalytic cycle of $CO_2$ hydrogenation reactions (**R1–R5**) with the model Ir(III)–PNP catalyst investigated in this work.

## 2. Materials and Methods

The theoretical background of URVA is provided in a comprehensive review article [55], therefore only some highlights are summarized. URVA explores the reaction complex (RC, i.e., the union of reacting molecules) along the path it traces out on the potential energy surface (PES) starting from the TS down into the entrance and exit channels, described by a large amplitude motion [56–59] as well as in the close vicinity of the reaction path, the so-called *reaction valley* being spanned by the vibrations perpendicular to the reaction path. [60,61]. The major focus of URVA is on the curvature of the reaction path. Since the reaction path is a curved line in $N_{vib}$ dimensional space with $N_{vib} = 3N - L$ internal coordinates ($L = 6$ for a non-linear $N$-atomic RC and 5 for an $N$-atomic linear RC), its direction and curvature can be derived with the Frenet-Serret formalism [62]. The reaction path direction at a path point $s$ is given by the unit vector $\boldsymbol{\eta}(s)$:

$$\boldsymbol{\eta}(s) = \frac{d\tilde{\mathbf{x}}(s)}{ds} = -\frac{\tilde{\mathbf{g}}(\tilde{\mathbf{x}}(s))}{c(s)} \tag{1}$$

where the derivative of the mass-weighted reaction coordinate $\tilde{\mathbf{x}}(s)$ with regard to $s$ is the normalized mass-weighted gradient vector $\tilde{\mathbf{g}}(s) \equiv \tilde{\mathbf{g}}(\tilde{\mathbf{x}}(s)) = \mathbf{M}^{1/2}\mathbf{g}(s)$ and $\mathbf{M}$ is a diagonal matrix of atomic masses. $c(s)$ is the normalization constant being equal to the length of the gradient vector $\|\tilde{\mathbf{g}}(s)\|$. The curvature vector $\boldsymbol{\kappa}(s)$ is given by [63,64]

$$\boldsymbol{\kappa}(s) = \frac{d^2\tilde{\mathbf{x}}(s)}{ds^2} = \frac{d\boldsymbol{\eta}(s)}{ds} = \frac{-1}{\|\tilde{\mathbf{g}}(s)\|}\left(\tilde{\mathbf{f}}^x(s)\boldsymbol{\eta}(s) - \left[\left(\boldsymbol{\eta}(s)\right)^{\dagger}\tilde{\mathbf{f}}^x(s)\boldsymbol{\eta}(s)\right]\boldsymbol{\eta}(s)\right) \tag{2}$$

where $\tilde{\mathbf{f}}^x(s)$ is the mass-weighted Hessian matrix in Cartesian coordinates $\mathbf{x}$.

Any electronic structure change, e.g., bond breaking/forming processes, but also rehybridization, charge polarization and transfer, etc. leads to distinct curvature peaks, which are directly reflected in the scalar reaction path curvature $\kappa(s) = \|\boldsymbol{\kappa}(s)\|$ in contrast to regions of minimal electronic change, which are reflected by curvature minima as sketched in Figure 2 showing the scalar curvature as a function of the reaction path parameter $s$. We call the region from one curvature minimum to the next embedding a curvature peak a *reaction phase*. Each chemical reaction has a unique pattern of curvature maxima and minima and as such a different number of reaction phases leading to a specific reaction

profile, the so-called *fingerprint* of the reaction [55]. Deeper insights into the nature of a chemical event reflected by a curvature peak are obtained from the decomposition of the scalar reaction curvature into individual components [65] such bond lengths, bond angles, and dihedral angles or other components such as puckering coordinates [66] denoted by the colored lines in Figure 2. Interesting to note is that even for complex catalytic reactions a curvature peak is most likely composed of a few components, which facilitates the analysis of the mechanistic details considerably. The sign of a component indicates if the parameter in question supports the chemical event (positive sign) or if it resists the chemical change (negative sign) [65]. For a detailed mathematical derivation, its connection to the work of Miller, Handy and Adams [67] and Kato and Morokuma [68] on the reaction path Hamiltonian, and recent advances of URVA, interested readers are referred to Refs [55,69]. Applications of URVA can be found, e.g., in Refs [69–84].

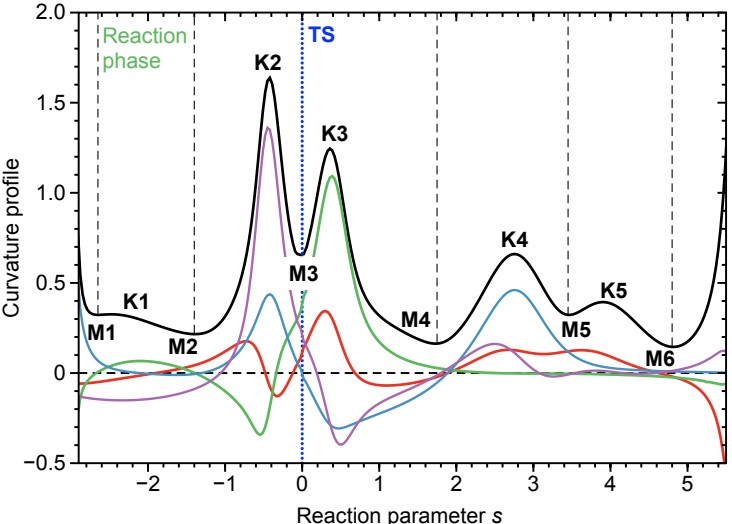

**Figure 2.** Schematic representation of the *reaction profile* for a model reaction defined as the scalar curvature $\kappa(s)$ (solid black line) given as a function of the reaction parameter $s$. Curvature minima $M$ and curvature maxima $K$ are shown. The location of the TS is denoted by a dotted blue line. Start and end of each reaction phase are denoted by dashed black lines. In addition the decomposition of the scalar curvature $\kappa(s)$ into four components is shown, blue, red, green and purple colored lines.

Similar as URVA, LMA is based on vibrational spectroscopy and it refines the use of normal vibrational force constants and frequencies derived from normal vibrational modes to characterize chemical bonds and/or weak chemical interactions. Normal vibrational modes of a polyatomic molecule are generally delocalized, as already stated by Wilson in 1941 via his proof that the associated normal mode coordinates are a linear combination of internal coordinates **q** or Cartesian coordinates **x** [85]. Therefore, normal mode stretching frequencies and associated stretching force constants are of limited use as individual bond strength descriptors. Konkoli and Cremer solved this problem via the transformation of normal vibrational modes into their local mode counterparts [86–90] (for details the reader may refer to two comprehensive review articles [91,92]). A local vibrational mode $\mathbf{a}_n$ is defined as

$$\mathbf{a}_n \;=\; \frac{\mathbf{K}^{-1}\mathbf{d}_n^\dagger}{\mathbf{d}_n\mathbf{K}^{-1}\mathbf{d}_n^\dagger} \tag{3}$$

implying the important result that all what is needed for the local mode analysis are the diagonal normal mode force constant matrix **K** in normal mode coordinates and the normal mode vectors $\mathbf{d}_n$ in internal coordinates, which can be obtained from a standard vibrational frequency calculation via the Wilson GF formalism [93–95], a routine part of most modern quantum chemistry packages.

Once the local mode vector $\mathbf{a}_n$ is known, one can define molecular properties corresponding to this local motion, such as local mode force constant, local mass and local frequency [86–90].

The corresponding local force constant $k_n^a$ of local mode $n$ (superscript $a$ denotes an adiabatically relaxed, i.e., local mode) [96] can be expressed as

$$k_n = \mathbf{a}_n^\dagger \mathbf{K} \mathbf{a}_n = \frac{1}{\mathbf{d}_n \mathbf{K}^{-1} \mathbf{d}_n^\dagger}. \tag{4}$$

As has been shown by Zou et al., local mode stretching force constants reflect the intrinsic strength of a chemical bond and/or weak chemical interaction [97].

The associated local mass is defined as

$$m^a = \frac{1}{\mathbf{B}_n \mathbf{M}^{-1} \mathbf{B}_n^\dagger} = \frac{1}{G_{nn}} \tag{5}$$

where $G_{nn}$ is a diagonal element of Wilson's inverse kinetic energy matrix $\mathbf{G}$ matrix and the Wilson $\mathbf{B}$ matrix provides the important relationship between internal and Cartesian coordinates via the first derivatives of the internal coordinates $q_n(n = 1, 2, 3 \ldots N_{vib})$ with regard to the Cartesian coordinates $x_i(i = 1, 2, 3 \ldots 3N)$ [93],

$$\mathbf{B}_n \quad = \quad \frac{\delta q_n(\mathbf{x})}{\delta x_i}. \tag{6}$$

From the local mode force constant $k^a$ and local mode mass $m^a$ the local mode frequency $\omega_n$ can be calculated

$$(\omega^a)^2 = 1/(4\pi^2 c^2)\frac{k_n^a}{m^a} = 1/(4\pi^2 c^2)G_{nn}\mathbf{a}_n^\dagger \mathbf{K} \mathbf{a}_n \tag{7}$$

with $c$ being the speed of light. In addition, the local mode infrared intensity has been defined which can be related to bond dipole moments [98].

LMA has been successfully applied to characterize covalent bonds [83,97,99–108] and weak chemical interactions such as hydrogen bonding [109–118], halogen bonding [119–125], pnicogen bonding [125–128], chalcogen bonding [107,125,129], tetrel bonding [130], metal bonding [115,121,131–136] as well as protein–ligand interactions [79,118,137,138].

It is convenient to transform local mode stretching force constants $k^a$ into relative bond strength order (BSO) via a power relationship $BSO = A * (k^a)^B$ being being based on the generalized Badger rule [83,139]. The dimensionless parameters A and B are obtained from two reference molecules with known BSO and $k^a$ values for the bond in question and the requirement that for a zero $k^a$ the BSO is zero too. We used different reference molecules for different bond types investigated in our study, for the Ir-ligand bonds we used Mayer's bond orders [140–142]. Details of the reference bonds and reference molecules are summarized in Table 1. The resulting A and B parameters for each bond are documented in the corresponding plots of the results and discussion part.

**Table 1.** Bond length R, local mode force constant $k^a$, bond strength order BSO, and energy density $H_\rho$ evaluated at the bond critical point $\rho(\mathbf{r})$ of the reference bonds and reference molecules used in our study. B3LYP/6-31G(d,p)/SDD(Ir) level of theory.

| Bond | R(Å) | $k^a$ (mDyn/Å) | BSO [1] | $H_\rho$ (Hr/Bohr$^3$) | Molecule |
|------|------|------|------|------|------|
| CO | 1.418 | 4.905 | 1.000 | −0.3719 | $CH_3OH$ |
| | 1.207 | 13.607 | 2.000 | −0.6883 | $CH_2O$ |
| CH, HH, OH | 1.150 | 1.203 | 0.500 | −0.1951 | $F_2H^{-1}$ |
| | 0.925 | 9.420 | 1.000 | −0.7275 | HF |
| IrO | 2.034 | 2.829 | 1.030 | −0.0300 | $Ir(CO)_5OH$ |
| | 1.822 | 5.340 | 1.469 | −0.1035 | $Ir(CO)_4O$ |
| IrH | 2.095 | 0.370 | 0.197 | −0.0110 | $Ir(CO)_5H$ |
| | 1.857 | 0.764 | 0.313 | −0.0219 | $Ir(CO)_5H_2$ |
| CO [2] | 1.169 | 16.404 | 2.271 | −0.7595 | $CO_2$ |
| HH [3] | 0.743 | 5.921 | 0.855 | −0.3360 | $H_2$ |

[1] The BSO values of bonds involving Ir atoms in the metal complexes are based on Mayer's bond orders [140–142].
[2] Based on the $CH_3OH$ and $CH_2O$ references. [3] Based on the on the HF and $F_2H^-$ references.

Bader's quantum theory of atoms-in-molecules (QTAIM) characterizes of chemical bonds based on the topological features of the total electron density $\rho(\mathbf{r})$ [143–146]. According to QTAIM a chemical bond between two atoms A and B is characterized by a bond path and a bond critical point $\rho$. The covalent character of the bond AB can be determined via the Cremer–Kraka criterion of covalent bonding [147–149]. According to this criterion, a chemical bond has covalent character when the energy density $H_\rho$, i.e., the energy density $H(\mathbf{r})$ taken at the at the bond critical point $\rho$ is negative, while a positive value of $H_\rho$ indicates an electrostatic character. $H(\mathbf{r}) = G(\mathbf{r}) + V(\mathbf{r})$; the positive kinetic energy density $G(\mathbf{r})$ describes electron density depletion and the negative potential energy density $V(\mathbf{r})$ describes electron density accumulation. Therefore, a chemical bond has covalent character when the potential energy density dominates.

We used in this study as a reaction path the intrinsic coordinate (IRC) path of Fukui [150,151] combined with the improved reaction path following procedure of Hratchian and Kraka, enabling the tracking of a chemical reaction far out into the entrance and exit channel [151]. We employed an IRC step size of 0.03 amu$^{1/2}$Bohr and an ultrafine grid for numerical DFT integrations. IRC calculations and the investigation of all stationary points were performed with the B3LYP density functional [152–154] utilizing Pople's 6-31G(d,p) basis set [155–157] and the Stuttgart–Dresden (SDD) effective core potential for iridium [158]. The energetics of the investigated chemical reactions was additionally recalculated via single energy point calculations with the coupled cluster DLPNO-CCSD(T) method [159] utilizing the def2-TZVP basis set [160] and the def2-ECP pseudo potential for iridium [158]. The DLPNO-CCSD(T) calculations were based on the DFT geometries and applying thermochemical corrections from the DFT frequency calculations. DFT calculations were performed with Gaussian program package [161], and DLPNO-CCSD(T) calculations with ORCA [162]. URVA was carried out using the program pURVA [163], and LMA analysis was performed using the LModeA program [164]. The Natural Bond Orbital (NBO) analysis [165–168] was applied for calculating NBO atomic charges along the reaction path. The QTAIM analysis was performed with the AIMALL program package [169].

### 3. Results and Discussion
#### 3.1. Energetics

Table 2 presents the activation and reaction energies $E^a$, $E_R$ and activation and reaction enthalpies $H^a$, $H_R$ of reactions **R0–R5** calculated at the B3LYP/6-31G(d,p)/SDD(Ir) and DLPNO-CCSD(T)/def2-TZVP/def2-ECP(Ir) levels of theory. The energy and enthalpy values in Table 2 are taken for each reaction relative to the energy of first point on the IRC (i.e., first point in the entrance channel corresponding to a van der Waals complex). The following discussion is based on the DLPNO-CCSD(T) results. According to Table 2,

the activation energy $E^a$ of the non-catalyzed reaction with a value of 76.3 kcal/mol is considerably larger then that for all reactions of the catalytic cycle, ranging between 1.7 kcal/mol for reaction **R4**, and 23.9 kcal/mol for reaction **R2**. Figure 3 displays the overall energetics of the catalytic cycle being composed of reactions **R1–R5**. All energies are relative to the energy of the van der Waals complex formed between the catalyst and $CO_2$ **1** in the entrance channel. All energies in Figure 3 are based on the DLPNO-CCSD(T) calculations performed at the end points of the DFT IRC calculations and for simplicity they have been calculated by adding the activation and reaction energy of the next reaction step of the cycle, to the product energy of the previous reaction step.

**Table 2.** The activation and reaction energy $E^a$, $E_R$ and the activation and reaction enthalpy $H^a$, $H_R$ of the reactions **R0–R5** investigated in this study [1]. The thermochemistry corrections of the DLPNO-CCSD(T) method is based on the DFT optimal geometry and frequency calculations. The B3LYP/6-31G(d,p)/SDD(Ir) and DLPNO-CCSD(T)/def2-TZVP/def2-ECP(Ir) levels of theory.

| | DFT | | | | DLPNO-CCSD(T) | | | |
|---|---|---|---|---|---|---|---|---|
| **Reaction** | $E^a$ | $E_R$ | $H^a$ | $H_R$ | $E^a$ | $E_R$ | $H^a$ | $H_R$ |
| **R0** | 72.2 | 3.6 | 71.8 | 9.2 | 76.3 | 5.1 | 75.9 | 10.7 |
| **R1** | 14.9 | −12.2 | 16.2 | −9.3 | 17.6 | −9.3 | 18.9 | −6.3 |
| **R2** | 23.9 | 22.3 | 22.8 | 21.7 | 23.9 | 22.9 | 22.7 | 22.3 |
| **R3** | 12.3 | 8.7 | 11.3 | 8.0 | 12.2 | 10.5 | 11.3 | 9.7 |
| **R4** | 1.2 | −10.3 | 0.6 | −8.5 | 1.7 | −13.6 | 1.2 | −11.7 |
| **R5** | 20.9 | −15.3 | 18.6 | −13.8 | 23.5 | −11.9 | 21.3 | −10.4 |

[1] Energy and enthalpy values (kcal/mol) from the end points of IRC and relative to the reactants of the corresponding reaction.

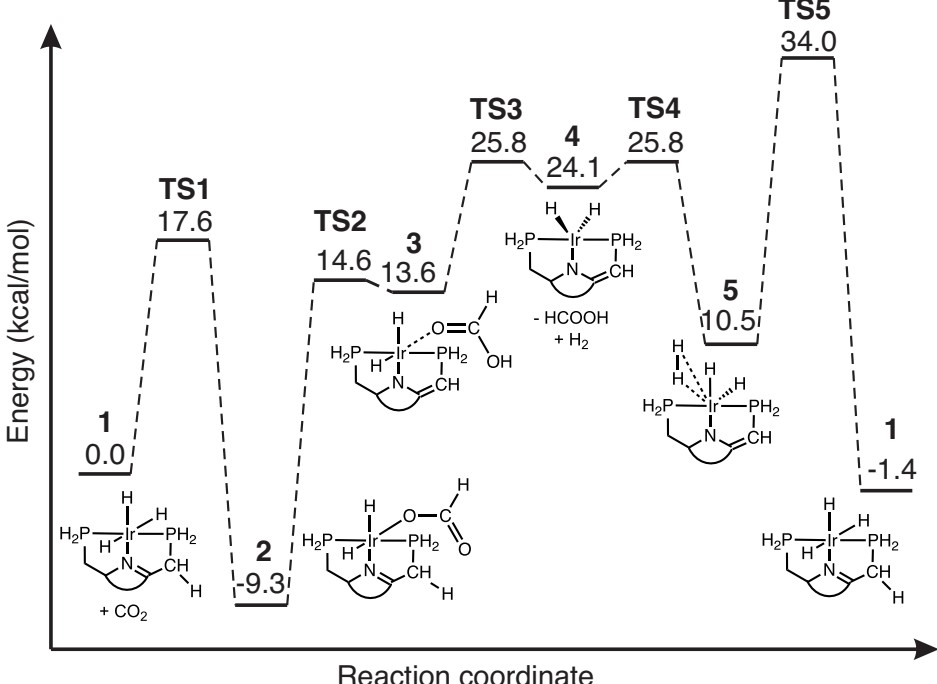

**Figure 3.** Energy diagram for the catalyzed reactions **R1–R5**, relative to the initial reactant van der Waals complex of **1**. For simplicity, the energies have been calculated by adding the activation and reaction energy of the next reaction step of the cycle, to the product energy of the previous reaction step. DLPNO-CCSD(T)/def2-TZVP/def2-ECP(Ir) level of theory performed at the end points of the DFT IRC calculations with B3LYP/6-31G(d,p)/SDD(Ir).

According to Table 2 and Figure 3, reactions **R1**—$CO_2$ addition ($E^a$ = 17.6 kcal/mol), **R2**—hydrogen transfer ($E^a$ = 23.9 kcal/mol), and **R5**—catalyst regeneration ($E^a$ = 23.5 kcal/mol),

have the largest activation energies, i.e., they are the decisive steps of the overall catalytic cycle. The values of the activation enthalpies $H^a$ for reactions **R1** (18.9 kcal/mol), **R2** (22.7 kcal/mol), and **R5** (21.3 kcal/mol) show a similar trend as seen from the activation energies $E^a$ of these reactions, which indicates small thermal energy contributions to the overall reaction energies. Although information about the energetics is useful, it does not disclose the actual mechanism taking place in each reaction step forming the catalytic cycle. This is explored in the next section.

### 3.2. Reaction Mechanism

*Reaction R0* Figure 4a presents the energy profile of the non-catalyzed reaction **R0** along the reaction path. The decomposition of the reaction curvature into selected components is presented in Figure 4b, selected geometrical parameters in Figure 4c and atomic NBO charges along the reaction path are presented in Figure 4d. According to Figure 4b, the reaction starts with initial formation of the $O_aH_a$ bond in reaction phase 2 (green line), with a resisting contribution of the $O_aC_aO_b$ bond angle (purple line). Bending of $CO_2$ molecule is also reflected in Figure 4c. The $O_aC_aO_b$ bond angle starts to decrease in phase 2. As shown in Figure 4a, both events the initial $O_aH_a$ bond formation and the $O_aC_aO_b$ bending, contribute 6.8 kcal/mol to the activation energy in this reaction phase. As revealed in Figure 4b, breaking of the $H_aH_b$ bond takes place in phase 4, which is confirmed by a strong supporting contribution of the $H_aH_b$ component (blue line). HH bond cleavage is also reflected by the change of $H_aH_b$ bond length (see Figure 4c), which starts to elongate in phase 4, making $H_a$ positively charged (see Figure 4d). $H_aH_b$ bond breaking is supported by the start of $C_aH_b$ bond formation (red line in Figure 4b), which is also confirmed by changes of $C_a$ and $H_b$ atomic charges (see Figure 4d). As shown in Figure 4a, the chemical events happening in phase 4, in particular the energy demanding breakage of the hydrogen bond, contributing 40.8 kcal/mol to the activation energy is the major cause of the high barrier. Finalization of $O_aH_a$ and $C_aH_b$ bonds takes place in phases 5 and 6 after TS (see Figure 4b),which is also reflected by $O_aH_a$ and $C_aH_b$ bond length changes (Figure 4c) and the NBO charge changes (Figure 4d). Obviously, the catalyzed reactions circumvents direct HH bond breakage, which is explored next.

*Reaction R1* Properties of reaction **R1** of the catalytic cycle along the reaction path are presented in Figure 5a–d. Reaction **R1** involves addition of $CO_2$ to the coordination sphere of iridium and hydrogen atom transfer from Ir to one of the $CO_2$ carbon atoms. According to Figure 5b the reaction starts in phase 3 with the initial stage of the $IrO_b$ bond formation (blue line) first resisting, followed by the next stage in phase 4 where it becomes supporting, whereas the final step of $IrO_b$ bond formation occurs after TS in phase 9. The different stages of IrO bond formation are in line with changes in $IrO_b$ bond lengths shown in Figure 5c. In phase 3 the $IrO_b$ bond distance starts to decrease, reaching its final value in phase 9. Figure 5d presents the corresponding changes of Ir and $O_b$ NBO charges taking place between reaction phases 4 and 9. The accompanying transfer of the H atom from Ir to the $C_a$ carbon of $CO_2$ implies $IrH_b$ bond cleavage and $C_aH_b$ bond formation. According to Figure 5b, the transfer process (red line) starts at the end of phase 4, where the formation of the new $C_aH_b$ bond (green line) is still resisting, changing into a large supporting contribution in phase 5.

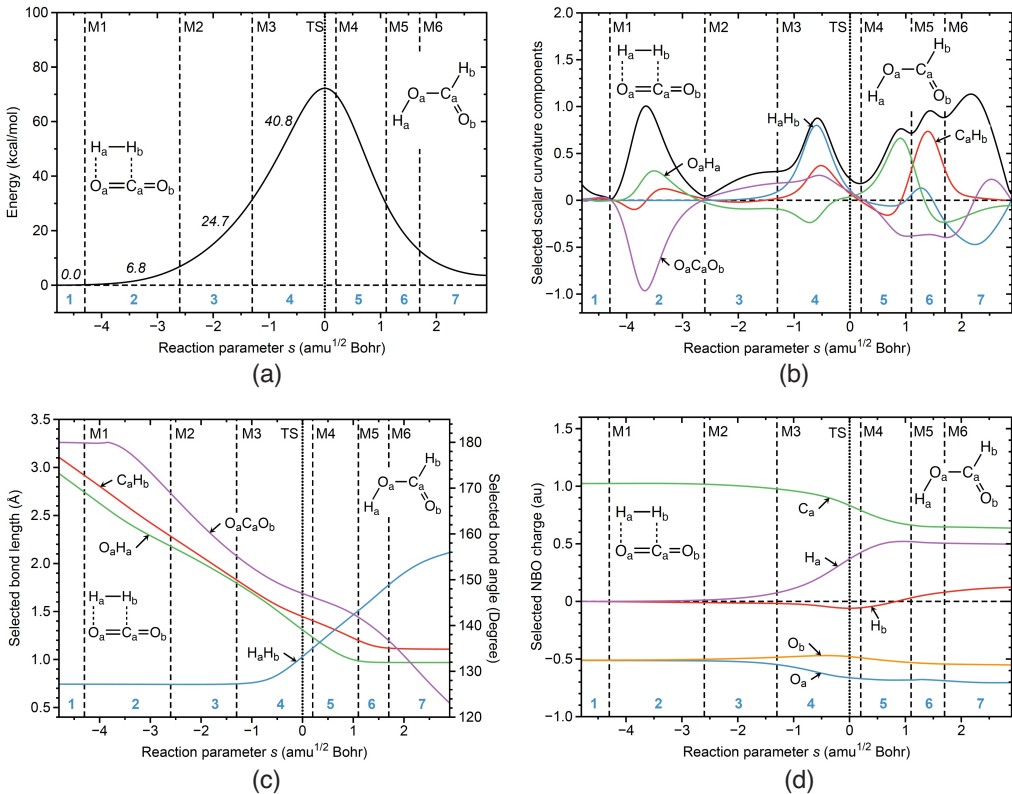

**Figure 4.** Properties of the gas phase reaction **R0** along the reaction path. (**a**) Energy profile, energy contributions of each reaction phase to the activation energy are indicated by italic numbers; (**b**) decomposition of the reaction curvature into selected components; (**c**) change of selected bond lengths along *s*; (**d**) change of selected NBO atomic charges along *s*. Positions of the curvature minima are shown as dashed vertical lines and are labeled as M1, M2, and so on. The position of the TS is indicated by a dotted line. Reaction phases are indicated by blue numbers. B3LYP/6-31G(d,p)/SDD(Ir) level of theory.

The next stage of $H_b$ transfer from Ir to $C_a$ occurs in phase 5, being finalized far out in the exit channel in phases 8–10. Figure 5c,d confirm these processes. Largest $C_aH_b$ bond length changes are found in phases 8 and 9 after the TS, as well as the changes in the $C_a$ and $H_b$ NBO charges. Figure 5b also reveals the involvement of the $O_aC_aO_b$ bond angle (purple line) in the reaction mechanism, with largest contribution between phases 4 and 9, in line with the OCO bond angle changes from 180° to 120° occurring between phases 4 and 9 (see Figure 5c) which is facilitated by the coordination of $C_b$ to the metal atom. According to Figure 5a, the two largest energy contributions to TS energy are 4.6 kcal/mol in phase 4, caused by $IrO_b$ and $C_aH_b$ bond formation, together with changes of the $O_aC_aO_b$ bond angle, and 6.2 kcal/mol in phase 8, connected with $IrH_b$ bond cleavage and $IrO_b$ bond formation, which in summary is far less than the direct cleavage of the HH bond in reaction **R0**.

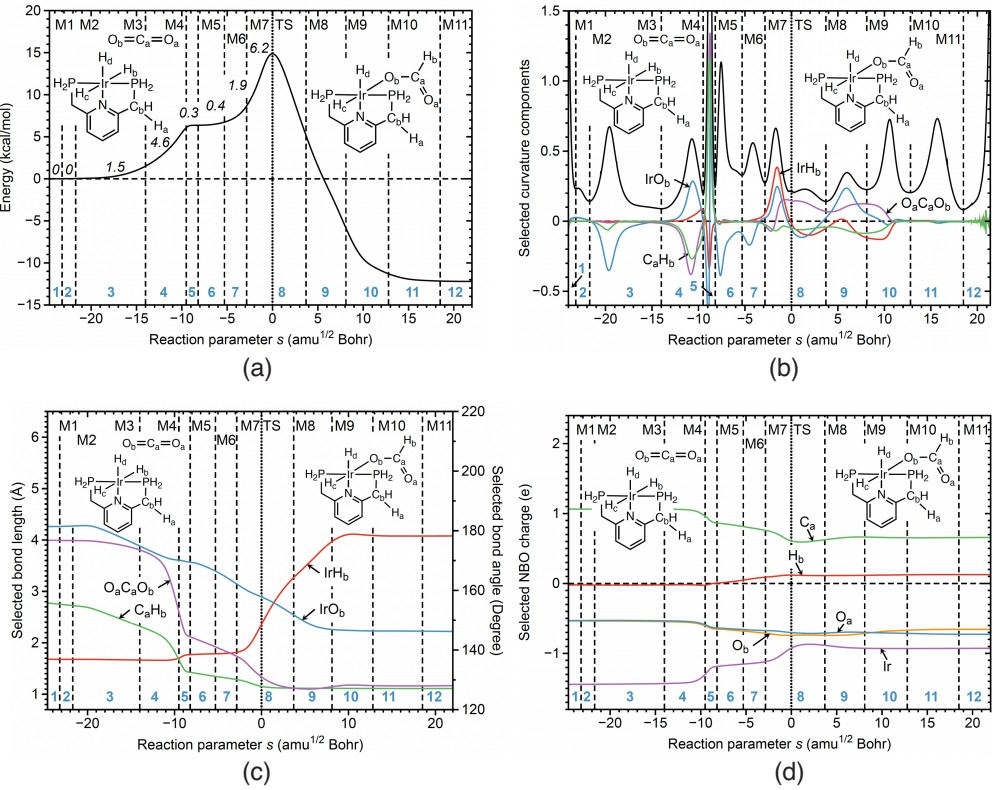

**Figure 5.** Properties of reaction **R1** of the catalytic cycle along the reaction path. (**a**) Energy profile, energy contributions of each reaction phase to the activation energy are indicated by italic numbers; (**b**) decomposition of the reaction curvature into selected components; (**c**) change of selected bond lengths along *s*; (**d**) change of selected NBO atomic charges along *s*. Positions of the curvature minima are shown as dashed vertical lines and are labeled as M1, M2, and so on. The position of the TS is indicated by a dotted line. Reaction phases are indicated by blue numbers. B3LYP/6-31G(d,p)/SDD(Ir) level of theory.

*Reaction R2* Reaction **R2** of the catalytic cycle is characterized by the H atom transfer from the $C_b$ carbon atom of the catalyst to the $O_a$ atom of $CO_2$. This is a necessary step for formation of the formic acid product, which goes along with elongation of the intermediate $IrO_b$ bond. Figure 6a–d show the corresponding URVA analysis. According to Figure 6b, $C_bH_a$ bond cleavage (orange line) takes place in phase 5, strongly supporting, and in phase 6, strongly resisting. The $C_bH_a$ distance starts to increase in phase 5, however it takes until the end of phase 10 to reach its final distance of 3.1 Å (see Figure 6c).

The formation of the new $O_aH_a$ bond starts in phase 4 (red line), as depicted in Figure 6b and is finalized in phase 6, also reflected by the corresponding change of the $O_aH_a$ distance shown in Figure 6c. What is intersting to note is that the changes of $O_aH_a$ and $C_bH_a$ distances complement each other. A secondary contribution to the mechanism is the change of $C_aO_a$ from double to single bond character (purple line in Figure 6b) in phases 5 and 6. Overall, changes in the atomic charges are only marginal (see Figure 6d). According to Figure 6a, the largest contributions to the activation energy are i) $O_aH_a$ bond formation and elongation of the $IrO_b$ bond in phase 4 (3.8 kcal/mol) and ii) hydrogen atom transfer accompanied by the change of the CO bond from double to single bond in phase 5 (9.8 kcal/mol).

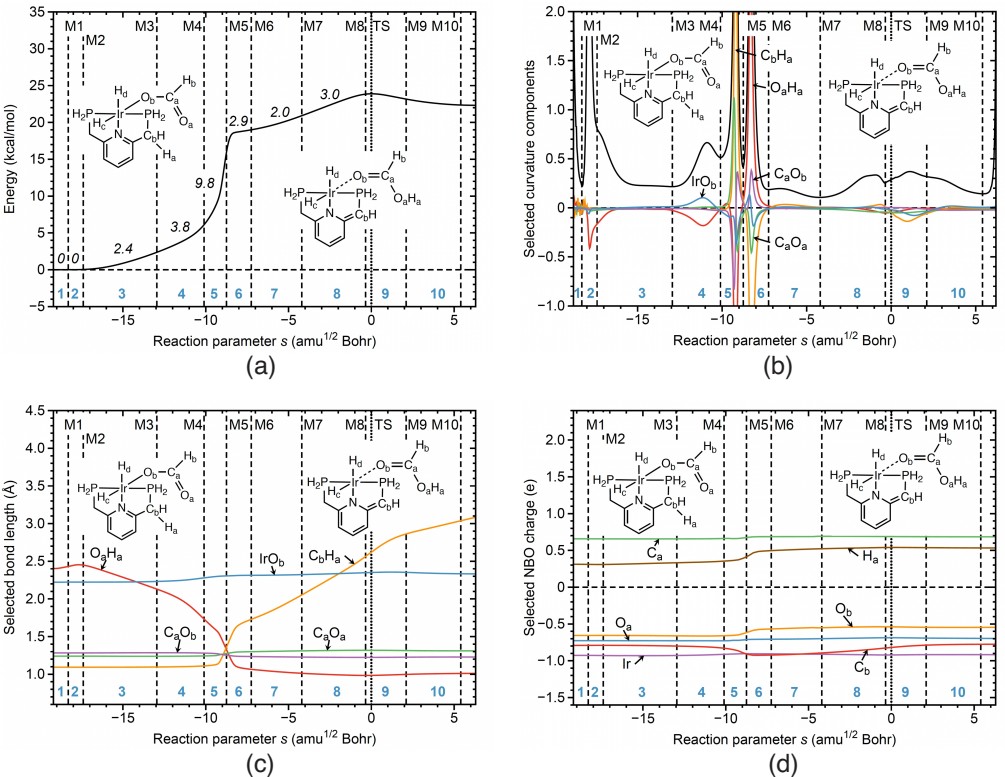

**Figure 6.** Properties of reaction **R2** of the catalytic cycle along the reaction path. (**a**) Energy profile, energy contributions of each reaction phase to the activation energy are indicated by italic numbers; (**b**) decomposition of the reaction curvature into selected components; (**c**) change of selected bond lengths along *s*; (**d**) change of selected NBO atomic charges along *s*. Positions of the curvature minima are shown as dashed vertical lines and are labeled as M1, M2, and so on. The position of the TS is indicated by a dotted line. Reaction phases are indicated by blue numbers. B3LYP/6-31G(d,p)/SDD(Ir) level of theory.

*Reaction R3* In reaction **R3** of the catalytic cycle, formic acid is released, accompanied by a cleavage of the intermediate $IrO_b$ bond (already weakened in **R2**), reducing the metal coordination number from 6 to 5, which leads to a reorganization the Ir coordination sphere, in particular a change in the $H_c$ and $H_d$ ligand positions. The URVA results of **R3** are summarized in Figure 7a–d. The curvature profile 7b shows that $IrO_b$ bond cleavage (blue line) starts in phase 2, with a supporting contribution. This is also reflected by a steep increase in the $IrO_b$ distance starting in that phase, (see Figure 7c) and an increase of the Ir charge (see Figure 7d). The next stages of $IrO_b$ bond cleavage occur in phases 3 and 4 (see $IrO_b$) first resisting and then supporting (phase 5, after the TS) with the final $IrO_b$ distance of 4.1 Å reached at the end of the reaction (see Figure 7c).

According to Figure 7b, there is also a contribution from the $NH_a$ hydrogen bond cleavage (pink line) in phase 3 (the $NH_a$ hydrogen bond is not shown in Figure 1 for simplicity), which starts in phase 2 and is finalized in phase 6. The cleavage of the $NH_a$ bond is also reflected in Figure 7c showing a steady increase of the NH bond distance from 1.75 Å to 3.2 Å. The change of the metal coordination sphere being related to the change of the $H_c$ and $H_d$ ligand positions is reflected by $IrH_c$ (orange line) and $IrH_d$ (red line) curvature components in phases 3 and 4. The contribution of the $IrH_d$ component is larger, indicating a stronger electronic structure change of the metal caused by displacement of this equatorial ligand. As shown in Figure 7a, the largest energy contribution of 7.0 kcal/mol to the activation energy results from the first stage of $IrO_b$ bond cleavage occurring in phase 2. The second stage of $IrO_b$ bond cleavage in phase 3, together with $IrH_c$ and $IrH_d$ ligand

reorganization and $NH_a$ hydrogen bond cleavage contribute 5.3 kcal/mol to the activation energy.

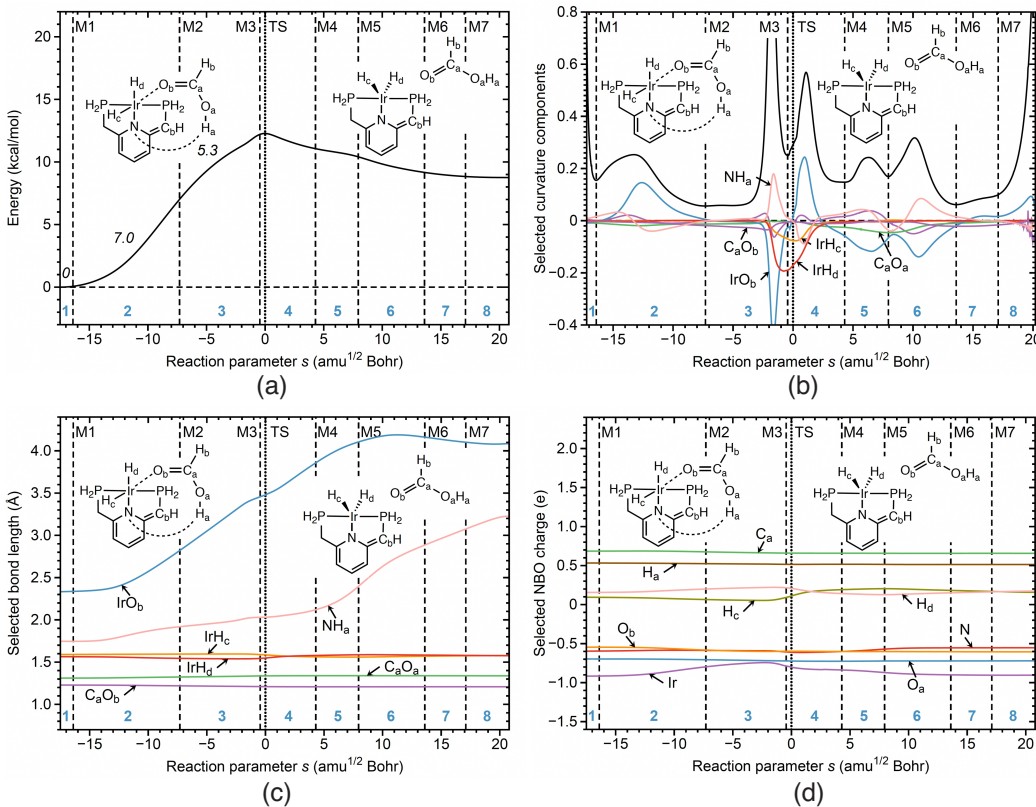

**Figure 7.** Properties of reaction **R3** of the catalytic cycle along the reaction path. (**a**) Energy profile, energy contributions of each reaction phase to the activation energy are indicated by italic numbers; (**b**) decomposition of the reaction curvature into selected components; (**c**) change of selected bond lengths along *s*; (**d**) change of selected NBO atomic charges along *s*. Positions of the curvature minima are shown as dashed vertical lines and are labeled as M1, M2, and so on. The position of the TS is indicated by a dotted line. Reaction phases are indicated by blue numbers. B3LYP/6-31G(d,p)/SDD(Ir) level of theory.

*Reaction R4* Figure 8a–d illustrate the change of the reaction properties along the reaction path for reaction **R4** of the catalytic cycle, which involves the addition of diatomic hydrogen to the coordination sphere of the metal via $\eta^2$–$H_2$ coordination; the first part of restoring the catalyst, which implies an increase of the Ir coordination number from 5 to 6. According to Figure 8b the $IrH_a$ (green line) and $IrH_b$ (blue line) bond components contribute to the reaction curvature in phases 2, 3, and 5. As depicted in Figure 8c $IrH_a$ and $IrH_b$ distances start to decrease in phase 2. Although the starting $H_a$ distance is 4.2 Å, considerably larger than the $H_b$ distance of 3.45 Å, both bonds are finalized simultaneously at the end of phase 5. There is also a minor contribution of the $H_aH_b$ component (red line) to the reaction curvature in phase 5 (see Figure 8b), reflecting changes of the $H_aH_b$ bond in this phase, where the $\eta^2$–$H_2$ complexation to the metal is finalized. The change of the metal coordination sphere in this reaction phase leads to an increase of negative charge on Ir atomic atom, as shown in Figure 8d. The activation energy of reaction **R4** is relatively small, (DFT result: $E^a$ = 1.2 kcal/mol, Figure 8a), mainly resulting from electronic structure reorganization of the $IrH_a$ and $IrH_b$ bonds taking place in phase 3.

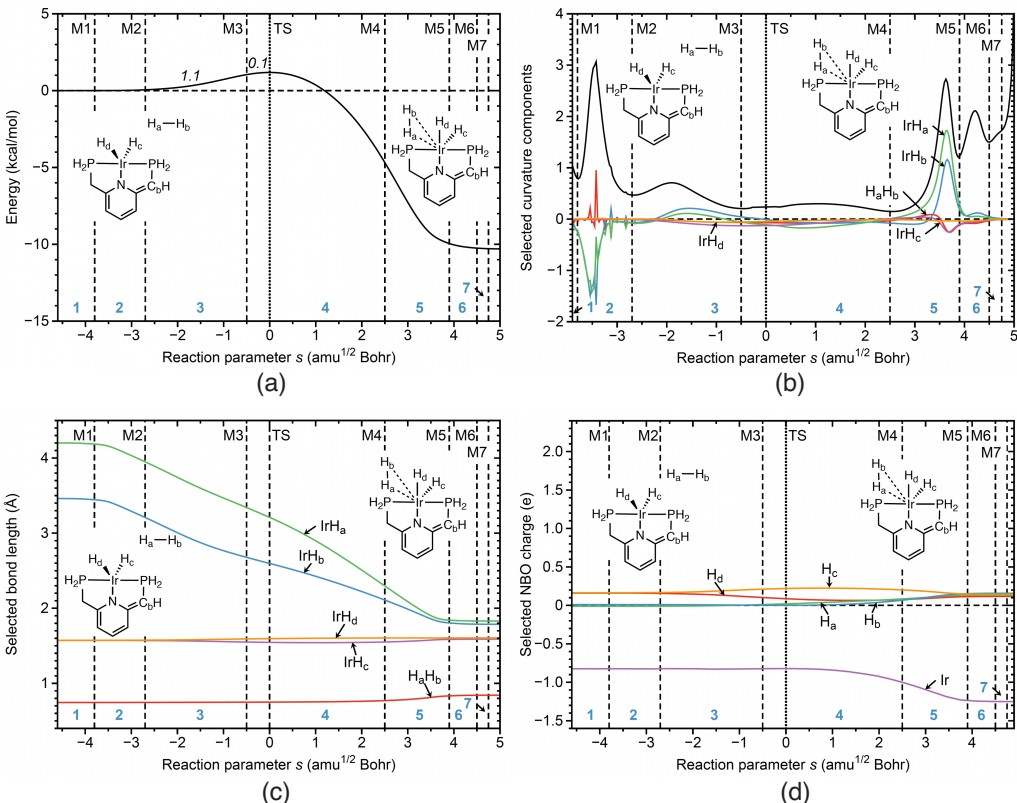

**Figure 8.** Properties of reaction **R4** of the catalytic cycle along the reaction path. (**a**) Energy profile, energy contributions of each reaction phase to the activation energy are indicated by italic numbers; (**b**) decomposition of the reaction curvature into selected components; (**c**) change of selected bond lengths along $s$; (**d**) change of selected NBO atomic charges along $s$. Positions of the curvature minima are shown as dashed vertical lines and are labeled as M1, M2, and so on. The position of the TS is indicated by a dotted line. Reaction phases are indicated by blue numbers. B3LYP/6-31G(d,p)/SDD(Ir) level of theory.

*Reaction R5* In reaction **R5** the initial catalyst is recovered, which implies $H_aH_b$ bond cleavage of the $\eta^2$–$H_2$ coordinated $H_2$ ligand, $IrH_a$ bond cleavage, migration of $H_a$ to $C_b$ and $C_bH_a$ bond formation. The corresponding URVA analysis is illustrated in Figure 9a–d. Figure 9b shows that $IrH_a$ bond cleavage (orange line) starts in phase 4 and proceeds through the following phases until phase 10. As reflected in Figure 9b, the initial $IrH_a$ distance of 1.8 Å starts to increase in phase 4 until its final value of 3.5 Å is reached at the end of the reaction. $H_aH_b$ bond cleavage (red line) starts at the border between phases 3 and 4, dominates phase 6 shortly before the TS with a large supporting contribution and proceeds parallel to $IrH_a$ bond cleavage until phase 10. It is interesting to note that elongation of the $H_aH_b$ bond starts in phase 6 (see Figure 9c) which parallels charge separation, $H_a$ migrating to the $C_b$ carbon atom to form a CH bond becomes more positively charged and $H_b$ forming a stronger IrH bond, more negatively charged (see Figure 9d). According to Figure 9b, $C_bH_a$ bond formation (green line) starts in phase 3 and is finalized in phase 7, after the TS and is revealed by a large curvature contribution.

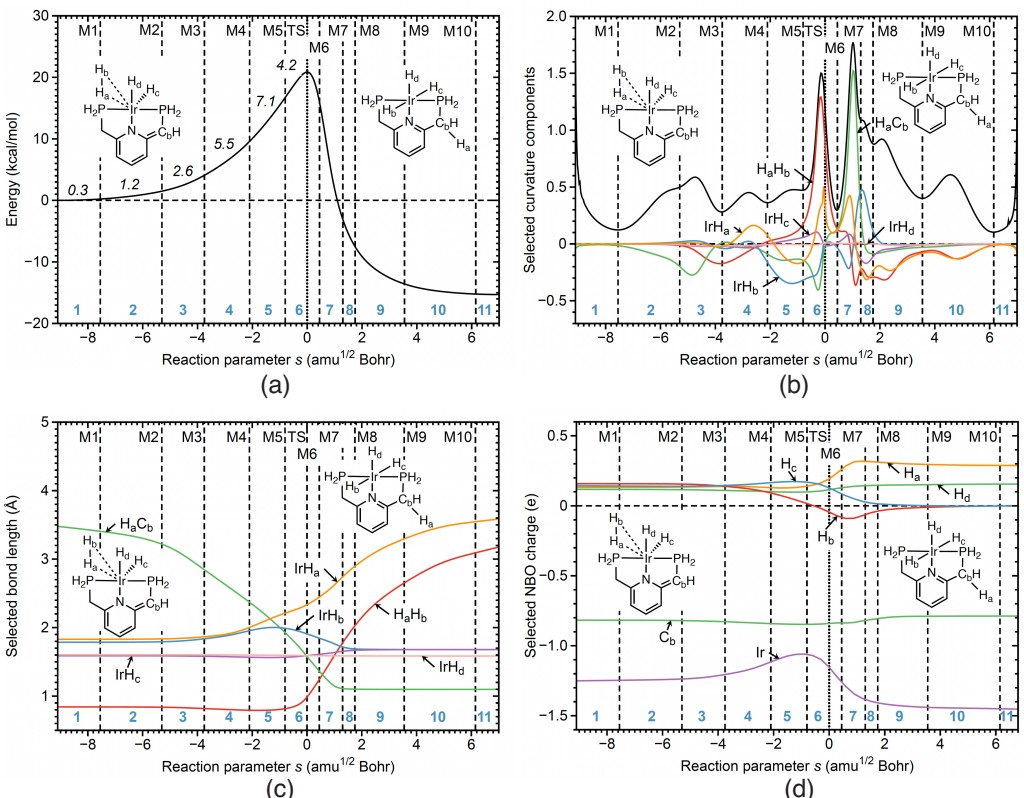

**Figure 9.** Properties of reaction **R5** of the catalytic cycle along the reaction path. (**a**) Energy profile, energy contributions of each reaction phase to the activation energy are indicated by italic numbers; (**b**) decomposition of the reaction curvature into selected components; (**c**) change of selected bond lengths along $s$; (**d**) change of selected NBO atomic charges along $s$. Positions of the curvature minima are shown as dashed vertical lines and are labeled as M1, M2, and so on. The position of the TS is indicated by a dotted line. Reaction phases are indicated by blue numbers. B3LYP/6-31G(d,p)/SDD(Ir) level of theory.

In phase 7 the $C_bH_a$ bond length reached its final value, as shown in Figure 9c. According to Figure 9a, the largest energy contributions to the activation energy occur in reaction phases 4 (5.5 kcal/mol) and 5 (7.1 kcal/mol), corresponding to $IrH_a$ bond cleavage and strengthening of the $IrH_b$ bond (see Figure 9b). In addition, it is important to note that $H_aHf_b$ bond cleavage contributes only 4.2 kcal/mol to the activation energy in the reaction phase 6, assisted by the charge polarization of both H atoms.

*3.3. Chemical Bond Analysis*

The URVA analysis was complemented with LMA and QTAIM bond analyses for all stationary points of reactions **R0–R5** in order provide further insides into the mechanism and to confirm the URVA analysis. Table 3 presents bond length R, local mode force constant $k^a$ and related bond strength order BSO and energy density $H_\rho$ for those bonds of reactants RE, transition states TS and products PR which change their character during the reaction. In order to limit the number of entries reported in this table, we assumed that the product of the previous catalyzed reaction is the same as the reactant of the next reaction in the catalytic cycle, although the IRC calculations did finish at slightly different geometries.

**Table 3.** Bond length R, local mode force constant $k^a$, bond strength order BSO, and energy density $H_\rho$ of the selected bonds for reactant RE, transition state TS, and product PR of reactions **R0–R5**. For the bond label description, see the text. B3LYP/6-31G(d,p)/SDD(Ir) level of theory.

| Bond | R (Å) | | | $k^a$ (mDyn/Å) | | | BSO | | | $H_\rho$ (Hr/Bohr³) | | |
|---|---|---|---|---|---|---|---|---|---|---|---|---|
| | RE | TS | PR | RE | TS | PR | RE | TS | PR | RE | TS | PR |
| **R0** | | | | | | | | | | | | |
| $C_aO_a$ | 1.169 | 1.260 | 1.353 | 16.364 | 10.621 | 5.972 | 2.267 | 1.690 | 1.143 | −0.7594 | −0.6537 | −0.4670 |
| $C_aO_b$ | 1.169 | 1.176 | 1.198 | 16.402 | 14.830 | 13.994 | 2.271 | 2.120 | 2.039 | −0.7597 | −0.7484 | −0.7315 |
| $C_aH_b$ | 3.128 | 1.447 | 1.108 | 0.019 | 0.865 | 4.805 | 0.124 | 0.447 | 0.797 | - | −0.0666 | −0.2896 |
| $O_aH_a$ | 2.959 | 1.308 | 0.969 | 0.017 | 1.686 | 8.005 | 0.119 | 0.560 | 0.947 | 0.0010 | −0.1021 | −0.5918 |
| $H_aH_b$ | 0.743 | 1.030 | 2.120 | 5.898 | 1.210 | 0.469 | 0.854 | 0.501 | 0.364 | −0.3364 | −0.0861 | - |
| **R1** | | | | | | | | | | | | |
| $C_aO_a$ | 1.170 | 1.242 | 1.240 | 16.098 | 9.679 | 10.562 | 2.242 | 1.587 | 1.684 | −0.7449 | −0.6444 | −0.6432 |
| $C_aO_b$ | 1.170 | 1.268 | 1.283 | 16.039 | 8.344 | 8.253 | 2.236 | 1.435 | 1.424 | −0.7436 | −0.6083 | −0.5787 |
| $C_aH_b$ | 2.769 | 1.153 | 1.111 | 0.041 | 3.090 | 4.692 | 0.160 | 0.687 | 0.791 | 0.0014 | −0.2259 | −0.2828 |
| $IrO_b$ | 4.262 | 2.893 | 2.222 | 0.049 | 0.415 | 1.184 | 0.107 | 0.352 | 0.633 | - | - | −0.0038 |
| $IrH_b$ | 1.679 | 2.359 | 4.079 | 1.983 | 0.415 | - | 0.576 | 0.212 | - | −0.0574 | −0.0003 | - |
| **R2** | | | | | | | | | | | | |
| $C_aO_a$ | 1.240 | 1.318 | 1.311 | 10.562 | 7.086 | 7.134 | 1.684 | 1.284 | 1.290 | −0.6432 | −0.5196 | −0.5320 |
| $C_aO_b$ | 1.283 | 1.224 | 1.229 | 8.253 | 11.827 | 11.316 | 1.424 | 1.818 | 1.765 | −0.5787 | −0.6731 | −0.6671 |
| $C_aH_b$ | 1.111 | 1.096 | 1.097 | 4.692 | 5.302 | 5.270 | 0.791 | 0.824 | 0.822 | −0.2828 | −0.3133 | −0.3115 |
| $C_bH_a$ | 1.094 | 2.623 | 3.084 | 5.237 | 0.138 | 0.117 | 0.821 | 0.241 | 0.228 | −0.2896 | - | - |
| $O_aH_a$ | 2.403 | 0.985 | 1.011 | 0.029 | 6.893 | 4.053 | 0.143 | 0.900 | 0.753 | 0.0020 | −0.5652 | −0.4934 |
| $IrO_b$ | 2.222 | 2.351 | 2.333 | 1.184 | 0.573 | 0.635 | 0.633 | 0.422 | 0.447 | −0.0038 | 0.0002 | −0.0002 |
| **R3** | | | | | | | | | | | | |
| $C_aO_a$ | 1.311 | 1.337 | 1.338 | 7.134 | 6.260 | 6.280 | 1.290 | 1.180 | 1.183 | −0.5320 | −0.4924 | −0.4907 |
| $C_aO_b$ | 1.229 | 1.210 | 1.209 | 11.316 | 12.883 | 13.069 | 1.765 | 1.927 | 1.946 | −0.6671 | −0.6978 | −0.6967 |
| $IrO_b$ | 2.333 | 3.476 | 4.085 | 0.635 | 0.076 | 0.050 | 0.447 | 0.136 | 0.108 | −0.0002 | 0.0004 | - |
| $IrH_c$ | 1.564 | 1.548 | 1.576 | 3.139 | 3.129 | 2.758 | 0.772 | 0.770 | 0.710 | −0.0981 | −0.1078 | −0.0949 |
| $IrH_d$ | 1.590 | 1.586 | 1.578 | 2.833 | 2.737 | 2.728 | 0.723 | 0.707 | 0.706 | −0.0874 | −0.0900 | −0.0939 |
| $NH_a$ | 1.747 | 2.031 | 3.225 | 0.219 | 0.129 | 0.029 | 0.282 | 0.236 | 0.143 | −0.0040 | −0.0007 | - |
| **R4** | | | | | | | | | | | | |
| $H_aH_b$ | 0.745 | 0.750 | 0.842 | 5.744 | 5.371 | 1.826 | 0.847 | 0.828 | 0.575 | −0.3330 | −0.3225 | −0.2020 |
| $IrH_a$ | 4.201 | 3.210 | 1.829 | 0.004 | 0.071 | 0.730 | 0.011 | 0.069 | 0.304 | - | - | - |
| $IrH_b$ | 3.461 | 2.597 | 1.787 | 0.004 | 0.050 | 0.807 | 0.011 | 0.055 | 0.324 | 0.0004 | 0.0001 | −0.0275 |
| $IrH_c$ | 1.576 | 1.545 | 1.589 | 2.758 | 3.318 | 2.783 | 0.710 | 0.799 | 0.715 | −0.0949 | −0.1091 | −0.0889 |
| $IrH_d$ | 1.578 | 1.595 | 1.604 | 2.728 | 2.760 | 2.762 | 0.706 | 0.711 | 0.711 | −0.0939 | −0.0859 | −0.0829 |
| **R5** | | | | | | | | | | | | |
| $H_aH_b$ | 0.842 | 0.986 | 3.182 | 1.826 | 0.707 | 0.037 | 0.575 | 0.418 | 0.155 | −0.2020 | −0.0907 | - |
| $IrH_a$ | 1.829 | 2.335 | 3.588 | 0.730 | 0.355 | 0.107 | 0.304 | 0.192 | 0.089 | - | - | - |
| $IrH_b$ | 1.787 | 1.909 | 1.679 | 0.807 | 0.567 | 1.987 | 0.324 | 0.259 | 0.576 | −0.0275 | −0.0202 | −0.0581 |
| $IrH_c$ | 1.589 | 1.591 | 1.679 | 2.783 | 2.737 | 1.988 | 0.715 | 0.707 | 0.576 | −0.0889 | −0.0863 | −0.0581 |
| $IrH_d$ | 1.604 | 1.592 | 1.585 | 2.762 | 2.842 | 2.914 | 0.711 | 0.724 | 0.736 | −0.0829 | −0.0869 | −0.0894 |
| $C_bH_a$ | 3.481 | 1.580 | 1.098 | 0.105 | 0.932 | 5.118 | 0.220 | 0.459 | 0.814 | - | −0.0371 | −0.2792 |

*Reaction R0* Reaction **R0** includes HH bond cleavage, formation of a new CH and OH bond, and the change of CO bond character in $CO_2$. There are five bonds involved in this reaction, namely $C_aO_a$, $C_aO_b$, $C_aH_b$, $O_aH_a$, and $H_aH_b$ (see Table 3). The $C_aO_a$ bond of $CO_2$ is changing its character from a double bond in RE to a single bond in PR, which is confirmed by the decreasing value of $k^a$ (16.364, 10.621, and 5.972 mDyn/Å for RE, TS and PR, respectively) which is in line with a bond length increase (1.169, 1.260, and 1.353 Å for RE, TS and PR, respectively) and the decreasing covalent character (−0.7594, −0.6537, and −0.4670, Hr/Bohr³ for RE, TS and PR, respectively). In contrast, the $C_aO_b$ bond of $CO_2$ keeps its double character during this reaction, however, its strength is slightly decreasing (16.402, 14.830, and 13.994 mDyn/Å, for RE, TS and PR, respectively), the bond length is slightly increasing (1.169, 1.176, and 1.198 Å for RE, TS and PR, respectively), and the covalent character is slightly decreasing (−0.7597, −0.7484, and −0.7315 Hr/Bohr³ for RE,

TS and PR, respectively). Two bonds are formed in this reaction, namely $C_aH_b$ and $O_aH_a$, while the $H_aH_b$ bond is cleaved. Consequently, the strength of the $C_aH_b$ bond is increasing (0.019, 0.865, and 4.805 mDyn/Å for RE, TS and PR, respectively), its length is decreasing (3.128, 1.447, and 1.108 Å for RE, TS and PR, respectively), and its covalent character is increasing ($-0.0666$ and $-0.2896$ Hr/Bohr$^3$, for TS and PR, respectively). Similarly, the strength of the $O_aH_a$ bond is increasing (0.017, 1.686, and 8.005 mDyn/Å for RE, TS and PR, respectively), its length is decreasing (2.959, 1.308, and 0.969 Å for RE, TS and PR, respectively), and its covalent character is increasing (0.0010, $-0.1021$, and $-0.5918$ Hr/Bohr$^3$ for RE, TS and PR, respectively). Finally, the $H_aH_b$ bond is cleaved in this reaction, which is confirmed by its decreasing strength (5.898, 1.210, and 0.469 mDyn/Å for RE, TS and PR, respectively), by its increasing length (0.743 1.030, and 2.120 Å for RE, TS and PR, respectively), and by its decreasing covalent character ($-0.3364$ and $-0.0861$ Hr/Bohr$^3$ for RE and TS, respectively). In summary, the LMA and QTAIM analyses taken at the stationary points are completely in line with the URVA analysis.

*Reaction R1* Reaction **R1** involves addition of $CO_2$ to the metal center forming the $IrO_b$ bond, H atom transfer from the metal to the $CO_2$ ligand via cleavage of the $IrH_b$ bond, and the formation of the $C_aH_b$ bond. Five bonds are involved in this process, namely $C_aO_a$, $C_aO_b$, $C_aH_b$, $IrO_b$, and $IrH_b$ (see Table 3). The $C_aO_a$ bond of $CO_2$ keeps more or less its double character during reaction **R1**, although its strength is somewhat decreasing as reflected by the $k^a$ values (16.098, 9.679, and 10.562 mDyn/Å for RE, TS and PR, respectively), increasing bond length (1.170, 1.242, and 1.240 Å for RE, TS and PR, respectively) and decreasing covalent character ($-0.7449$, $-0.6444$, $-0.6432$ Hr/Bohr$^3$ for RE, TS and PR, respectively). In contrast, the $C_aO_b$ bond of $CO_2$ is changing its character from a double to a single bond upon $CO_2$ addition to the metal. The strength of the $C_aO_b$ bond is decreasing (16.039, 8.344, and 8.253 mDyn/Å for RE, TS and PR, respectively), the bond length is increasing (1.170 1.268, and 1.283 Å for RE, TS and PR, respectively) and the covalent character is decreasing ($-0.7436$, $-0.6083$, and $-0.5787$ Hr/Bohr$^3$ for RE, TS and PR, respectively). The changes of the $C_aO_b$ bond are a consequences of the $IrO_b$ bond formation in this reaction. The strength of the $IrO_b$ bond is consequently increasing ($k^a = 0.049$, 0.415, and 1.184 mDyn/Å for RE, TS and PR, respectively) and its length is decreasing (4.262, 2.893, and 2.222 Å for RE, TS and PR, respectively). We observed only for the PR $IrO_b$ bond a bond critical with a small covalent character ($-0.0038$ Hr/Bohr$^3$). The reaction **R1** also involves a H atom transfer from the catalyst to the incoming $CO_2$ ligand, where the $IrH_b$ bond of the catalyst is cleaved, and the new $C_aH_b$ bond is formed. Consequently, the strength of the $IrH_b$ bond is decreasing ($k^a = 1.983$ and 0.415 mDyn/Å for RE and TS, respectively), its length is increasing (1.679, 2.359, and 4.079 Å for RE, TS and PR, respectively), and its covalent character is decreasing ($-0.0574$ and $-0.0003$ Hr/Bohr$^3$ for RE and TS, respectively). The formation of the $C_aH_b$ bond is reflected by increasing bond strength (0.041, 3.090, and 4.692 mDyn/Å for RE, TS and PR, respectively), decreasing bond length (2.769 1.153, and 1.111 Å for RE, TS and PR, respectively) and increasing covalent character (0.0014, $-0.2259$, and $-0.2828$ Hr/Bohr$^3$ for RE, TS and PR, respectively). In conclusion, we observe that the results of the LMA and QTAIM analyses of the reaction **R1** are in support of the results of the URVA analysis, similarly as for the reaction **R0**.

*Reaction R2* Reaction **R2** involves H atom transfer from the catalyst to the $CO_2$ ligand and subsequent OH bond formation implying the cleavage of a catalyst CH bond. There are six bonds involved in this process, $C_aO_a$, $C_aO_b$, $C_aH_b$, $C_bH_a$, $O_aH_a$, and $IrO_b$ (see Table 3). As the $H_a$ atom is transferred from $C_b$ to the $CO_2$ ligand, the strength of the $C_bH_a$ bond is decreasing ($k^a = 5.237$, 0.138, and 0.117 mDyn/Å for RE, TS and PR, respectively), and its length is consequently increasing (1.094, 2.623, and 3.084 Å for RE, TS and PR, respectively). We observed only a bond critical point for the RE $C_bH_a$ bond with a relatively large covalent character ($-0.2896$ Hr/Bohr$^3$). The change of $O_aH_a$ bond properties are going into the opposite direction. The $O_aH_a$ bond strength is increasing ($k^a = 0.029$, 6.893, and 4.053 mDyn/Å for RE, TS and PR, respectively), the bond length is decreasing (2.403,

0.985, and 1.011 Å for RE, TS and PR, respectively), and the covalent character is increasing (0.0020, −0.5652, and −0.4934 Hr/Bohr$^3$ for RE, TS and PR, respectively). It is interesting to note that at the TS the $O_aH_a$ bond is stronger (6.893 mDyn/Å), shorter (0.985 Å) and more covalent (−0.5652 Hr/Bohr$^3$) than in the PR. As the reaction proceeds, the $C_aO_a$ bond is changing its character from double bond to a single bond, which is reflected in the RE, TS, and PR bond properties. The strength of the $C_aO_a$ bond is decreasing ($k^a$ = 10.562, 7.086, and 7.134 mDyn/Å for RE, TS and PR, respectively), its length is increasing (1.240, 1.318, and 1.311 Å for RE, TS and PR, respectively), and its covalent character is decreasing (−0.6432, −0.5196, and −0.5320 Hr/Bohr$^3$ for RE, TS and PR, respectively). $C_aO_b$ bond properties are going in the opposite direction, as the character of this bond is changing from a single to a double bond. The strength of the $C_aO_b$ bond is increasing ($k^a$ = 8.253, 11.827, and 11.316 mDyn/Å for RE, TS and PR, respectively), its length is decreasing (1.283 1.224, and 1.229 Å RE, TS and PR, respectively), and its covalent character is increasing (−0.5787, −0.6731, and −0.6671 Hr/Bohr$^3$ for RE, TS and PR, respectively). The change of the $C_aO_b$ bond properties, affects in turn the properties of the $IrO_b$ bond. The $IrO_b$ bond becomes weaker ($k^a$ = 1.184, 0.573, and 0.635 mDyn/Å for RE, TS and PR, respectively), longer (2.222, 2.351, and 2.333 Å for RE, TS and PR, respectively), and less covalent (−0.0038, 0.0002, and −0.0002 Hr/Bohr$^3$ for RE, TS and PR, respectively). The strength of the $C_aH_b$ bond increases only slightly, as reflected by its $k^a$ values (4.692, 5.302, and 5.270 mDyn/Å for RE, TS and PR, respectively), its bond length (1.111, 1.096, and 1.097 Å for RE, TS and PR, respectively), and its covalent character (−0.2828, −0.3133, and −0.3115 Hr/Bohr$^3$ for RE, TS and PR, respectively). Again, these results are in line with the URVA analysis.

*Reaction R3* Reaction **R3** involves the dissociation of the HCOOH product from the metal coordination sphere via $IrO_b$ bond cleavage, affecting also the properties of both $C_aO_a$ and $C_aO_b$ bonds. In addition, this process changes the metal coordination numbers from 6 to 5, which is accompanied by the change of $H_c$ and $H_d$ atomic positions and the change of $IrH_c$ and $IrH_d$ bond properties. As such, six bonds are involved, $C_aO_a$, $C_aO_b$, $IrO_b$, $IrH_c$, $IrH_d$, and $NH_a$ (see Table 3). The $IrO_b$ bond strength is decreasing ($k^a$ = 0.635, 0.076, and 0.050 mDyn/Å for RE, TS and PR, respectively), the bond becomes longer (2.333, 3.476, and 4.085 Å for RE, TS and PR, respectively) and less covalent (−0.0002 and 0.0004 Hr/Bohr$^3$ for RE and TS , respectively; no bond critical point was found for PR). Cleavage of the $IrO_b$ bond affects the properties of the $C_aO_b$ bond, which becomes stronger ($k^a$ = 11.316, 12.883, and 13.069 mDyn/Å for RE, TS and PR, respectively), shorter (1.229, 1.210, and 1.209 Å for RE, TS and PR, respectively), and more covalent (−0.6671, −0.6978, and −0.6967 Hr/Bohr$^3$ for RE, TS and PR, respectively). In contrast, the $C_aO_a$ bond becomes weaker ($k^a$ = 7.134, 6.260, and 6.280 mDyn/Å for RE, TS and PR, respectively), longer (1.311, 1.337, and 1.338 Å for RE, TS and PR, respectively) and less covalent (−0.5320, −0.4924, and −0.4907 Hr/Bohr$^3$ for RE, TS and PR, respectively), although the overall changes of $C_aO_a$ bond properties are smaller than those for the $C_aO_b$ bond. Change of the metal coordination number from 6 to 5 during this reaction implies a change of the $H_c$ and $H_d$ ligand positions in the metal coordination sphere, which affects the $IrH_c$ and $IrH_d$ bond properties. The $IrH_c$ bond becomes weaker (3.139, 3.129, and 2.758 mDyn/Å for RE, TS and PR, respectively), longer (1.564, 1.548, and 1.576 Å for RE, TS and PR, respectively), and less covalent (−0.0981, −0.1078, and −0.0949 Hr/Bohr$^3$ for RE, TS and PR, respectively). However, although the $IrH_d$ bond becomes somewhat weaker ($k^a$ = 2.833, 2.737, and 2.728 mDyn/Å for RE, TS and PR, respectively), it becomes shorter (1.590, 1.586, and 1.578 Å for RE, TS and PR, respectively), and slightly more covalent (−0.0874, −0.0900, and −0.0939 Hr/Bohr$^3$ for RE, TS and PR, respectively). The values of the $IrH_c$ and $IrH_d$ PR bond properties are similar ($k^a$ = 2.758 and 2.728 mDyn/Å, R = 1.576 and 1.578 Å, and $H_\rho$ = −0.0949 and −0.0939 Hr/Bohr$^3$, for $IrH_c$ and $IrH_d$, respectively), which is consistent with their similar positions in the metal coordination sphere. There is an additional NH bond involved in reaction **R3** with hydrogen bond character $N \cdots H_aO_a$, which is cleaved during the reaction. The strength of the $NH_a$ bond continuously decreases ($k^a$ = 0.219, 0.129, and 0.029 mDyn/Å for RE, TS and PR, respectively), the length increases (1.747, 2.031, and 3.225 Å for RE, TS and PR,

respectively), and the covalent character decreases ($-0.0040$ and $-0.0007$ Hr/Bohr$^3$ for RE and TS R, respectively, no bond critical point was found for PR. In conclusion, the results of the LMA and QTAIM analyses are consistent with the results of the URVA analysis.

*Reaction R4* Reaction **R4** involves the $\eta^2$–$H_2$ coordination of $H_2$ to the metal center, the first step of catalyst recovery. The coordination number of the metal is changed back from 5 to 6 and two new bonds are formed between the metal and $H_2$, i.e., $IrH_a$ and $IrH_b$, changing the properties of the $H_aH_b$ bond from the incoming $H_2$ ligand. $H_2$ addition also changes properties of the $IrH_c$ and $IrH_d$ bonds caused by the change of the $H_c$ and $H_d$ ligand positions in the coordination sphere. Five bonds are involved in this process, $H_aH_b$, $IrH_a$, $IrH_b$, $IrH_c$ and $IrH_d$ (see Table 3). The strength of the $IrH_a$ bond is increasing ($k^a = 0.004$, $0.071$, and $0.730$ mDyn/Å) for RE, TS and PR, respectively) and its length is decreasing ($4.201$, $3.210$, and $1.829$ Å for RE, TS and PR, respectively). No bond critical point could be found for any of the three stationary points. Similarly, the strength of the $IrH_b$ bond is increasing as well ($k^a = 0.004$, $0.050$, and $0.807$ mDyn/Å for RE, TS and PR, respectively), the length is decreasing ($3.461$ $2.597$, and $1.787$ mDyn/Å for RE, TS and PR, respectively), and its covalent character is increasing ($0.0004$, $0.0001$, and $-0.0275$ Hr/Bohr$^3$ for RE, TS and PR, respectively). The $H_2$ addition to the metal changes properties of the $H_aH_b$ bond, which becomes weaker ($k^a = 5.744$, $5.371$, and $1.826$ mDyn/Å for RE, TS and PR, respectively), longer ($0.745$ $0.750$, and $0.842$ Å for RE, TS and PR, respectively) and less covalent ($-0.3330$, $-0.3225$, and $-0.2020$ Hr/Bohr$^3$ for RE, TS and PR, respectively). This is important for the final step of the catalytic cycle, in which the $H_aH_b$ bond has to be cleaved. The $H_c$ and $H_d$ atoms are moving into new positions, affecting the $IrH_c$ bond which becomes slightly stronger ($k^a = 2.758$, $3.318$, and $2.783$ mDyn/Å for RE, TS and PR, respectively), slightly longer ($1.576$, $1.545$, and $1.589$ Å for RE, TS and PR, respectively), and less covalent ($-0.0949$, $-0.1091$, and $-0.0889$ Hr/Bohr$^3$ for RE, TS and PR, respectively). Similar changes were found for the $IrH_d$ bond. The bond strength is increasing ($k^a = 2.728$, $2.760$, and $2.762$ mDyn/Å for RE, TS and PR, respectively), the bond length is increasing ($1.578$, $1.595$, and $1.604$ Å for RE, TS and PR, respectively), and the covalent character is decreasing ($-0.0939$, $-0.0859$, and $-0.0829$ Hr/Bohr$^3$ for RE, TS and PR, respectively). These results fully support the URVA analysis.

*Reaction R5* The last reaction of the catalytic cycle involves the dissociation of the $H_aH_b$ bond from the $H_2$ ligand, the transfer of the $H_a$ atom from the metal center to the $C_b$ carbon atom of the catalyst accompanied by the cleavage of the $IrH_a$ bond, and the strengthening of the two $IrH_b$ and $C_bH_a$ bonds, regenerating the original catalyst. Six bonds are involved in this process, $H_aH_b$, $IrH_a$, $IrH_b$, $IrH_c$, $IrH_d$ and $C_bH_a$ (see Table 3). The strength of the already weakened $H_aH_b$ bond decreases further ($k^a = 1.826$, $0.707$, and $0.037$ mDyn/Å for RE, TS and PR, respectively), its length increases ($0.842$, $0.986$, and $3.182$ Å for RE, TS and PR, respectively), and its covalent character decreases ($-0.2020$ and $-0.0907$ Hr/Bohr$^3$ for RE and TS, respectively, no bond critical point was found for PR). The strength of the $IrH_a$ bond is decreasing ($k^a = 0.730$, $0.355$, and $0.107$ mDyn/Å or for RE, TS and PR, respectively) and its length is increasing ($1.829$, $2.335$, and $3.588$ Å for RE, TS and PR, respectively) without bond critical points in this reaction. $C_bH_a$ bond formation is confirmed by increasing bond strength ($k^a = 0.105$, $0.932$, and $5.118$ mDyn/Å for RE, TS and PR, respectively)), decreasing bond length ($3.481$, $1.580$, and $1.098$ Å for RE, TS and PR, respectively)) and increasing covalent character ($-0.0371$ and $-0.2792$ Hr/Bohr$^3$ for TS and PR, respectively, no bond critical point was found for RE). Strengthening of the $IrH_b$ bond is reflected by increasing $k^a$ values ( $0.807$, $0.567$, and $1.987$ mDyn/Å or RE, TS and PR, respectively), decreasing bond length ($1.787$, $1.909$, and $1.679$ Å or RE, TS and PR, respectively), and increasing covalent character ($-0.0275$, $-0.0202$, and $-0.0581$ Hr/Bohr$^3$ or RE, TS and PR, respectively). The strength of the $IrH_c$ bond is decreasing ($k^a = 2.783$, $2.737$, and $1.988$ mDyn/Å for RE, TS and PR, respectively), the length is increasing ($1.589$, $1.591$, and $1.679$ Å for RE, TS and PR, respectively), and the covalent character is decreasing ($-0.0889$, $-0.0863$, and $-0.0581$ Hr/Bohr$^3$ for RE, TS and PR, respectively) whereas the properties of the $IrH_d$ bond

are changing in the opposite direction. The strength of this bond increases ($k^a$ = 2.762, 2.842, and 2.914 mDyn/Å for RE, TS and PR, respectively), the length decreases (1.604, 1.592, and 1.585 Å for RE, TS and PR, respectively), and the covalent character increases (−0.0829, −0.0869, and −0.0894 Hr/Bohr$^3$ RE, TS and PR, respectively). Again, these results fully support the URVA analysis.

In conclusion, the LMA and QTAIM analyses performed at all stationary points of reactions **R0**–**R5** deepen the insights obtained with URVA by adding bond specific details. In the last section, overall trends of the LMA and QTAIM bond properties based on Table 3 data are presented to round up the results and discussion part.

*3.4. Comparison of Relative Bond Strength Orders BSO*

Local mode force constants $k^a$ can be transformed into relative bond strength orders BSO, which asses bond strength in a more commonly used language referring to single and multiple bond character of a bond, and in this way offering a more convenient way to compare bond strengths in aseries of molecules. BSO values were obtained described in the computational section, based on the reference molecules summarized in Table 1. Figure 10a–d show BSO values as a function of the local mode force constants $k^a$, for the CO, CH, HH, and OH bonds and Figure 11a–b for IrO and IrH bonds. There are two types of CO bonds, $C_aO_a$ and $C_aO_b$, which are changing their bond strengths in reactions **R0**–**R3**. According to Figure 10a, largest CO BSO values are in the range of 2.2 indicating strong double bond character. The strongest strongest CO bonds were found for the RE of the non-catalyzed reaction **R0** with BSO values close to the BSO of 2.271 of the CO bond in the $CO_2$ molecule (see Table 1). The smallest BSO values are in the range of 1.2 indicating single bond character, with the TS values in between. The weakest CO bond is found again for the PR of the non-catalyzed reaction **R0**.

Figure 10b shows the BSO values for $C_aH_b$ and $C_bH_a$ bonds of the catalyst, which are changing in reactions **R0**–**R2**, and **R5**. For example, the dissociating $C_bH_a$ bond of reaction **R2** starts with an RE BSO value of 0.821, which is close to a single bond bond strength and drops down in the PR to a value of 0.228, indicating a weak CH interaction, already shown in the TS with a BSO value of 0.241. The BSO values for the $H_aH_b$ bonds are presented in Figure 10c for reactions **R0**, **R4**, and **R5**. The HH bond of the RE for both non-catalyzed reaction **R0** and catalyzed reaction **R4** are the strongest with BSO values in the range of 0.85, which are close to the BSO value of 0.855 of the HH bond of the free $H_2$ molecule (see Table 1. The weakest HH interaction is found for the RE of reaction **R5** with a BSO value of 0.575, facilitating the cleavage of this bond. There is one OH formed in reactions **R0** and **R2**. The small RE BSO value of 0.119 reflects that there is only a weak interaction in the RE van der Waals complex. The BSO value increases to 0.560 at the TS and the PR BSO of 0.947 indicates OH single bond character upon OH bond formation. Figure 11a–b present plots of the BSO values for the IrO and IrH bonds. The $IrO_b$ bonds change in reactions **R1**–**R3**. Corresponding BSO values are presented in Figure 11a. For example, in reaction **R1** the $IrO_b$ bond is formed, starting with a small RE BSO value of 0.107, which increases to 0.352 for the TS and to 0.633 for the PR. It is interesting to note that the overall BSO values of all IrO bonds in this study are smaller than 1, indicating relatively weak bond strength character. Figure 11b shows the BSO values of the IrH bonds, such as $IrH_a$, $IrH_b$, $IrH_c$, and $IrH_d$, which are formed, cleaved, and changed in reactions **R1**, and **R3**– **R5**. For example, in reaction **R3**, upon dissociation of the HCOOH ligand, the $IrH_c$ bond is only changing slightly its character, with a BSO values of 0.772 for the RE, 0.770 for the TS, and 0.710 for the PR.

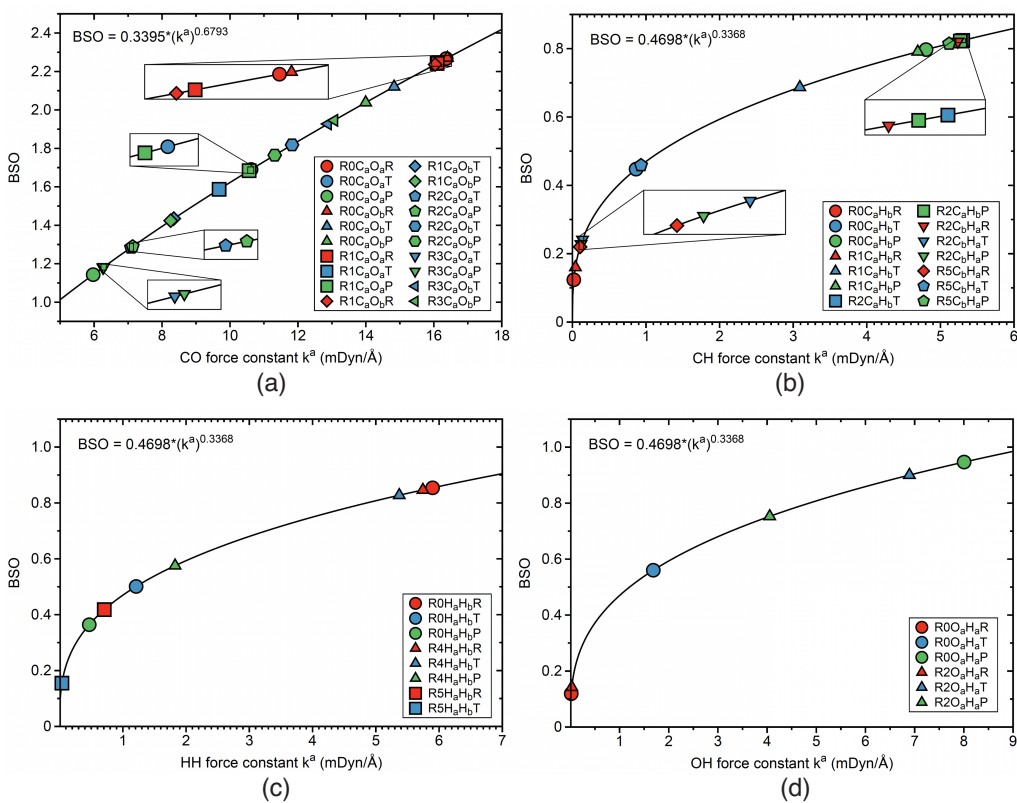

**Figure 10.** BSO values as a function of the local mode force constants $k^a$, calculated from the power relationship $BSO = A * (k^a)^B$; for details, see text and Table 1. (**a**) CO bonds; (**b**) CH bonds; (**c**) HH bonds; (**d**) OH bonds. Labels $R0C_aO_aR$, $R0C_aO_aT$, and $R0C_aO_aP$ indicate reactant RE (red color), TS (blue color), and product PR (green color) of the bond involving the $C_a$ and $O_a$ atoms of reaction **R0**. Similarly notation for the reactions **R1**–**R5**. For the bond label description see the text. B3LYP/6-31G(d,p)/SDD(Ir) level of theory.

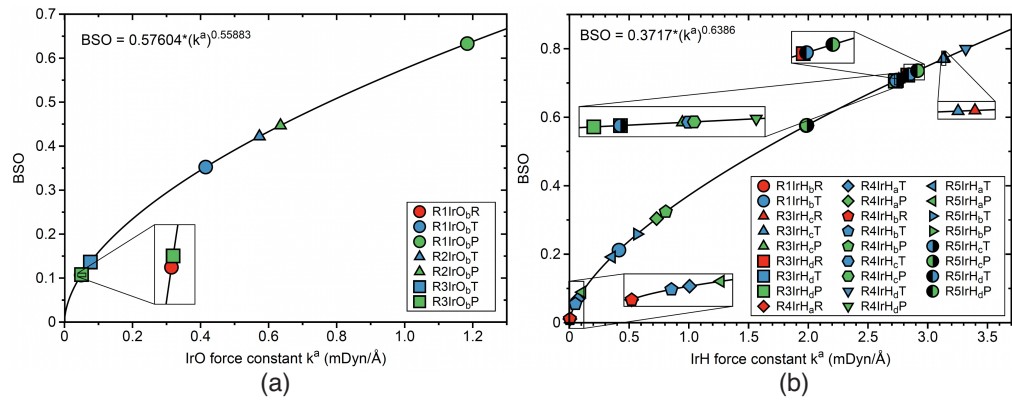

**Figure 11.** BSO values as a function of the local mode force constants $k^a$, calculated from the power relationship $BSO = A * (k^a)^B$; for details, see text and Table 1. (**a**) IrO bonds; (**b**) IrH bonds. Labels $R1IrO_bR$, $R1IrO_bT$, and $R1IrO_bO$ indicate reactant RE (red color), TS (blue color), and product PR (green color) of the bond involving the Ir and $O_b$ atoms of thereaction **R1**. Similarly notation for reactions **R2**–**R5**. For the bond label description see the text. B3LYP/6-31G(d,p)/SDD(Ir) level of theory.

### 3.5. Correlations between Bond Properties and Local Mode Force Constants

Figures 12 and 13 present the correlation between bond strength expressed by the local mode force constants and bond length, which according to the Badger rule [139,170] leads to a power relationship. Figure 12a shows the correlation for CO bonds of reactions **R0–R3**.

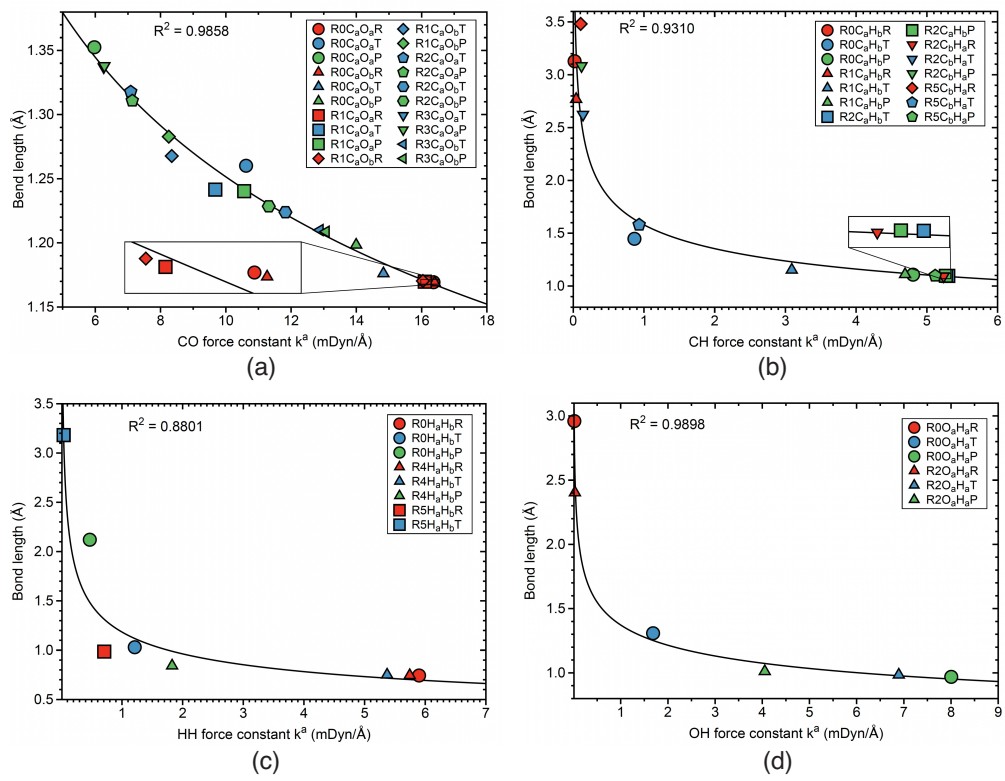

**Figure 12.** Correlation between bond lengths and local mode force constants. (**a**) CO bonds; (**b**) CH bonds; (**c**) HH bonds; (**d**) OH bonds. Labels such as $R0C_aO_aR$, $R0C_aO_aT$, and $R0C_aO_aP$ indicate reactant RE (red color), TS (blue color), and product PR (green color) $C_aO_a$ bond in reaction **R0**. Similarly notation for reactions **R1–R5**. For the bond label description see the text. B3LYP/6-31G(d,p)/SDD(Ir) level of theory.

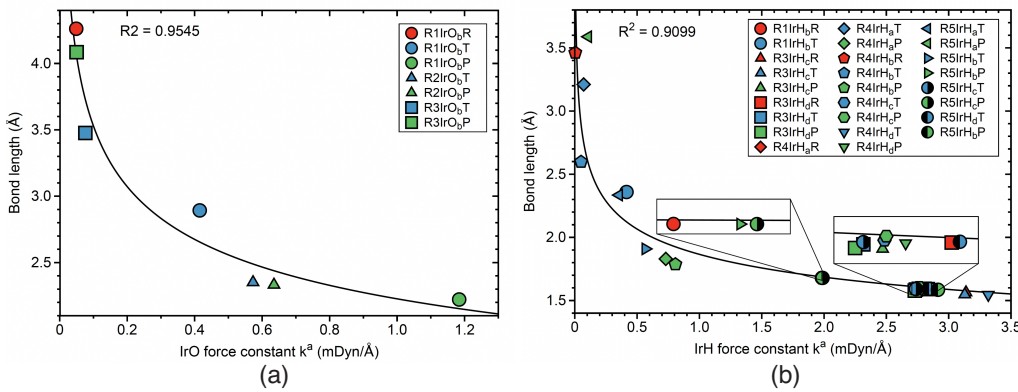

**Figure 13.** Correlation between bond length and the local mode force constants. (**a**) IrO bon; (**b**) IrH bonds. Labels such as $R1IrO_bR$, $R1IrO_bT$, and $R1IrO_bO$ indicate reactant RE (red color), TS (blue color), and product PR (green color) $IrO_b$ bond in reaction **R1**. Similarly notation for reactions **R2–R5**. For the bond label description see the text. B3LYP/6-31G(d,p)/SDD(Ir) level of theory.

According to Figure 12a, stronger CO bonds are connected to shorter CO bonds. The $R^2$ value of 0.9858 confirms a significant correlation between these two properties. larger

divergencies are observed for the $C_aO_a$ bond at TS of reaction **R0**, and $C_aO_a$ and $C_aO_b$ bonds at the TSs of reaction **R1**. Figure 12b shows a similar correlation between bond length and bond strength for the CH bonds in reactions **R0–R2**, and **R5**. The $R^2$ factor of this correlation has a somewhat lower value of 0.9310, corresponding to a good correlation. The divergency occurs for the $C_aH_b$ bond at the TS of reaction **R0**. The correlation between bond strength and bond length for the HH bonds are presented in Figure 12c, showing the results for the $H_aH_b$ bonds in reactions **R0**, **R4**, and **R5**. This correlation has an $R^2$ value of 0.8801, with outliers for PRs of reactions **R0** and **R4** and the RE of reaction **R5**. Figure 12d shows a significant correlation between bond strength and bond length for the $O_aH_a$ bonds of reactions **R0** and **R2**. The value of the $R^2$ factor of this correlation is 0.9898 with no substantial outlier. The bond strength bond versus length correlation for the $IrO_b$ bonds in reactions **R1–R3** is presented in Figure 13a. The $R^2$ factor of this correlation has a moderate value 0.9545, with the largest outlier for the TSs of reactions **R1** and **R2**. Figure 13b shows the correlation between bond strength and bond length for the IrH bonds of reactions **R1**, and **R3–R5**. The $R^2$ value of 0.9099 indicates a moderate correlation, smaller than that found for the IrO bonds. Largest outliers are found for the $IrH_b$ TS bond of reaction **R1**, the $IrH_a$ and $IrH_b$ TS bonds of reactions **R5**, and for the PR $IrH_a$ and $IrH_c$ bonds of reaction **R4**. In summary, we observed a satisfactory correlation between bond strength and bond length, in line with the popular rule that the stronger bond is also the shorter bond. However, this is not always the case as demonstrated by the increasing number of exceptions from this simple rule, reported in the literature [87,88,90,171,172]. The outliers in the correlations presented in this work are mostly related to the TS stationary points. At the TS, in particular for bond breaking/forming processes, significant electronic structure rearrangements take place, e.g., leading to a stage between two different hybridization forms, which can cause an unusual bonding situation that cannot be captured by such an empirical correlation.

*Energy density versus Local mode force constant* As outlined above, the energy density $H_\rho$ is convenient measure of the covalent character of a chemical bond or weak chemical interaction based on the electron density. Overall, we observed in our study that the stronger bonds have more covalent character, expressed by a more negative value of $H_\rho$. Figures 14 and 15 present the correlations between $H_\rho$ and $k^a$ based on a linear relationship. Figure 14a shows the correlation for the $C_aO_a$ and $C_aO_b$ bonds of reactions **R0–R3**. The $R^2$ factor has a value of 0.9411, with a number of small outliers mostly for TSs. The bond strength- energy density correlation for the $C_aH_b$ and $C_bH_a$ bonds of reactions **R0–R2**, is presented in Figure 14b. The $R^2$ value of 0.9758 reflects a good correlation, with the largest outlier for the TS $C_aH_b$ bond of reaction **R1**. Figure 14c shows the same correlation for the $H_aH_b$ bonds of reactions **R0**, **R4**, and **R5**, with a $R^2$ value of 0.9438. The largest outlier was found for the PR $H_aH_b$ bond of reaction **R4**. The correlation between energy density and bond strength for the $O_aH_a$ bonds of reactions **R0** and **R2** is shown in Figure 14d. The $R^2$ factor for this reaction, has a value of 0.9256 reflecting a moderate correlation with the largest outlier for the PR $O_aH_a$ bond of reaction **R2**. Figure 15a presents the correlation between $H_\rho$ and $k^a$ for the $IrO_b$ bonds in reactions **R1–R3**. As obvious from the $R^2$ factor of 0.7816, there is only some trend between these two properties. One also has to consider the scarcity of the data points. It is interesting to note that the PR $IrO_b$ bond of reaction **R2** has an $H_\rho$ close to zero, indicating a more electrostatic character of this bond. A similar situation occurs for the TS $IrO_b$ bond of reaction **R3**. A better correlation between energy density and bond strength is observed for the IrH bonds of reactions **R1** and **R3–R5** shown in Figure 15b. The $R^2$ factor of that correlation has a value of 0.9829, with some outlier such as the TS $IrH_b$ bond of the reaction **R1**, or the TS $IrH_c$ bond of reaction **R3**.

In summary, overall, we observed a good correlation between energy density $H_\rho$ and bond strength expressed by the local mode force constant $k^a$, which indicates that the strength of the bonds investigated in this work is predominantly based on their covalent character. However, a caveat is appropriate. $H_\rho$ reflects bonding at just one point of the electron density, namely the bond critical point, whereas the $k^a$ as a second order property picks up the electronic environment [91]. As such it is more sensitive in the case of complex

bonding situation as, e.g., found for the IrO bonds showing only a trend but not a significant correlation between these two properties.

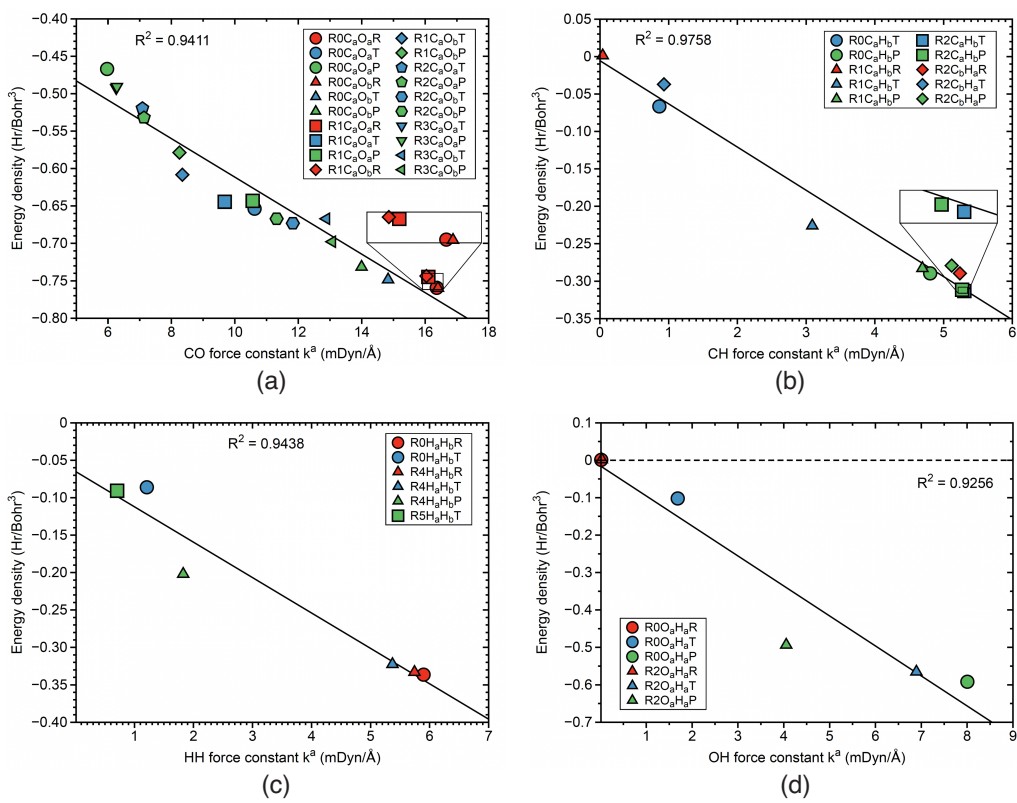

**Figure 14.** Correlation between energy density $H_\rho$ and local mode force constant $k^a$. (**a**) CO bonds; (**b**) CH bonds; (**c**) HH bonds; (**d**) OH bonds. Labels such as R0$C_aO_a$R, R0$C_aO_a$T, and R0$C_aO_a$P indicate reactant RE (red color), TS (blue color), and product PR (green color) $C_aO_a$ in reaction **R0**. Similar notation for reactions **R1–R5**. For the bond label description see the text. B3LYP/6-31G(d,p)/SDD(Ir) level of theory.

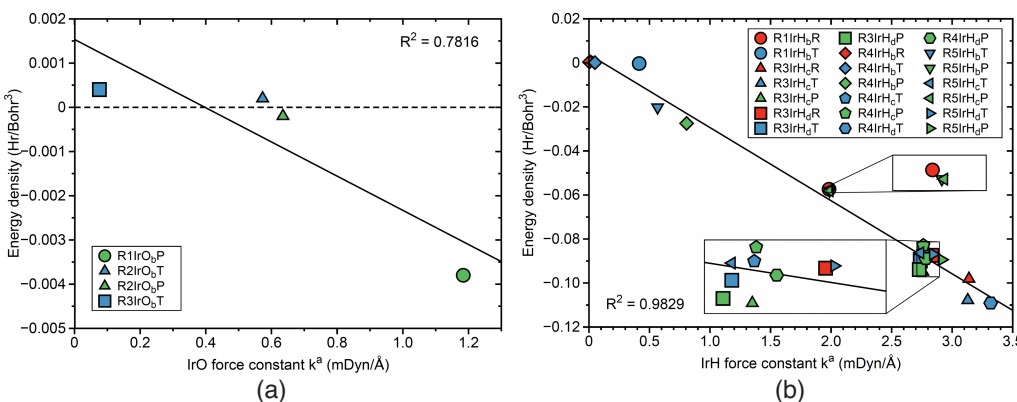

**Figure 15.** Correlation between energy density $H_\rho$ and local mode force constant $k^a$. (**a**) IrO bonds; (**b**) IrH bonds. label such as R1IrO$_b$R, R1IrO$_b$T, and R1IrO$_b$O indicate reactant RE (red color), TS (blue color), and product PR (green color) IrO$_b$ bond in **R1**. Similar notation for reactions **R2–R5**. For the bond label description see the text. B3LYP/6-31G(d,p)/SDD(Ir) level of theory.

## 4. Conclusions and Outlook

We investigated in this study the catalytic cycle (reactions **R1–R5**) of carbon dioxide hydrogenation catalyzed with an iridium pincer model complex, which reduces the unfavorable activation energy of 76 kcal/mol for the non-catalyzed gas phase reaction **R0**

by more than 100%. As a computational tool we used URVA to explore the mechanistic features of each reaction step in all detail, shedding new light on which are the most energy consuming events in the non-catalyzed reaction, and how are they avoided in the catalyzed reaction. URVA was complemented by LMA and QTAIM analyses at all stationary points of each reaction. We used the DFT level of theory for URVA, LMA, and QTAIM, whereas for the energetics a dual approach (i.e., single point DLPNO-CCSD(T) energy calculations for all stationary points based on the DFT geometries) was applied. The most important findings of our study showing for the first time how the catalyst effectively manages to keep energy demand at a balance are summarized in the following list:

- According to the URVA analysis, the most energy consuming process of the non-catalyzed reaction **R0** is the cleavage of the HH bond of the $H_2$ reactant, which takes place before TS, and as such contributes to the activation energy with 40.8 kcal/mol.
- In order to avoid direct HH bond cleavage, the catalyst divides the one–step non-catalytic reaction into a multistep catalytic cycle; $CO_2$ addition to the catalyst, H atom transfer between catalyst and $CO_2$ ligand, release of HCOOH product, addition of $H_2$, reorganization of the coordination sphere of the catalyst to achieve its original form. Each of these decisive steps could be clearly identified in the characteristic curvature profiles for the first time.
- The expensive cleavage of the HH bond in the non-catalyzed reaction, is replaced in the catalyzed reaction by H transfer (reaction **R2**) requiring the cleavage of an IrH bond with a significantly smaller contribution of 9.8 kcal/mol to the activation energy, which is revealed by both energy and curvature profiles.
- The dissociation of the final product from the catalyst (reaction **R3**) is characterized by the cleavage of an IrO bond and an intermediate NH hydrogen bond, and according to our URVA analysis, both contribute to the activation energy with a moderate amount of 12.3 kcal/mol.
- As unravelled by the URVA curvature profiles most of the events related to the reorganization of the catalyst to restore its original form (reaction **R5**) occur after the TS, i.e., they do not contribute to the activation energy.

In summary, the URVA analysis complimented with the LMA and QTAIM methods provided new comprehensive mechanistic details of all reaction steps forming the catalytic cycle of the hydrogenation of carbon dioxide, which hopefully will stimulate and inspire the community working on this important topic. Experiments suggest that the catalytic activity of iridium pincer complexes for the hydrogenation of carbon dioxide can be considerably increased under basic reaction conditions [43,44]. A follow-up URVA study is planned to explore this.

**Supplementary Materials:** The following supporting information can be downloaded at: https://www.mdpi.com/article/10.3390/inorganics10120234/s1, containing reaction movies **R0–R5** and the Cartesian coordinates of the stationary point, i.e, the reactant complex, TS and product complex for all six reactions **R0–R5**.

**Author Contributions:** Conceptualization, E.K. and M.F.; methodology, M.F. and E.K.; validation, E.K. and M.F.; formal analysis, M.F. and E.K.; investigation, M.F. and E.K.; resources, E.K.; data curation, M.F.; writing—original draft preparation, M.F.; writing—review and editing, E.K.; funding acquisition, E.K. All authors have read and agreed to the published version of the manuscript.

**Funding:** This research was funded by the National Science Foundation NSF, grant CHE 2102461.

**Data Availability Statement:** All data supporting the results of this work are presented in tables and figure of the manuscript and in the Supplementary Materials.

**Acknowledgments:** We thank Juliana Antonio for many useful comments and suggestions. This work was supported by the National Science Foundation, Grant CHE 2102461. We thank the Center for Research Computation at SMU for providing generous high-performance computational resources.

**Conflicts of Interest:** The authors declare no conflict of interest.

## Abbreviations

The following abbreviations are used in this manuscript:

| | |
|---|---|
| URVA | unified reaction valley approach |
| DFT | density functional theory |
| DLPNO-CCSD(T) | domain–based local pair natural orbital of coupled cluster single and double with perturbative triple excitations |
| PNP | 2,6–bis(di–isopropylphosphinomethyl)pyridine |
| TOF | turnover frequency |
| TON | turnover number |
| LMA | Local Modes Analysis |
| PES | potential energy surface |
| RC | reaction complex |
| QTAIM | quantum theory of atoms-in-molecules |
| IRC | intrinsic reaction coordinate |
| NBO | natural bond orbital |
| BSO | bond strength order |
| Hr | hartree |
| RE | reactant |
| TS | transition state |
| PR | product |

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
