# Peer review of "URVA and Local Mode Analysis of an Iridium Pincer Complex Efficiently Catalyzing the Hydrogenation of Carbon Dioxide"

_inorganics, doi:10.3390/inorganics10120234_

Round 1

Reviewer 1 Report

This manuscript describes a very detailed analysis of the hydrogenation of carbon dioxide catalyzed by an iridium pincer complex. The level of theory (B3LYP/6-31G(d,p)/SSD) is appropriate. Single point calculations with DLPNO-CCSD(T) give very similar energies to the DFT calculations. The unified reaction valley approach (URVA) was used to examine the changes in structure and bonding for the uncatalyzed reaction and the five steps in the catalyzed reaction.  Local mode analysis and QTAIM were also used to characterize the changes in structure and bonding and were found to be consistent with the URVA analysis.  The manuscript is suitable for publication after minor revisions, as described below.

((a)    Figure 1. Add a ChemDraw structure of the full catalyst (i.e. with the isopropyl groups). Replacing them with hydrogens in the calculation should be fine.

((b)    Eq (1) – identify c(s)

((c)     Figure 2. What is the vertical blue line in phase 1

((d)    A little heavy on the self-citations (69-84, 97-138)

((e)    Figure 3. How does the energy profile change if you consider free energies rather than enthalpies? Relative rates depend on the free energies of activation. There is an extra 10-15 kcal/mol in bimolecular reactions that comes from the change in entropy. Does this change the rate determining step?

((f)      Table 1. In the units for the energy density, does Hr stand for hartree? Perhaps better to spell it out that to use what seems to be a nonstandard abbreviation.

((g)    ChemDraw structures in the plots are very useful for understanding the phases of each reaction step in the catalytic cycle.  Please add some additional figures with 3D structures of the transition states and intermediates so that a reader can better follow the geometric transformations that occur (and additional key structures along the reaction path that correspond to features on the energy profile such as phase 5 of R1, M5 of R2, etc.). Without such pictures the paper would not be understandable by the general inorganic audience.

((h)    The SI lists movie files but they were not accessible in the SI provided to the reviewer. The website listed for the SI was not functional.

((i)      Reaction R2 discussion. I presume the first sentence should read: “from the Cb carbon atom of the catalyst to the Oa atom of CO2”

((j)      Reaction R3. Need to see a structure showing the NH hydrogen bond.

((k)    The analysis of the changes in structure and bonding in the catalytic cycle is very, very detailed and the three different methods (URVA, LMA, QTAIM) are in good accord.  The conclusion lists a number of points that are easy to deduce from the mechanism even without this analysis. What new insights are gained from the analysis that were not already apparent from the mechanism and from previous computational work?  Highlighting these new insights in the conclusion would justify the detailed analysis and hopefully stimulate other researchers to use the URVA approach.

Reviewer 2 Report

The article is well written and could be accepted for publication in “Inorganics”. This work comprises a significant addition to the chemistry of pincer complexes as catalyst for the hydrogenation of carbon dioxide. I recommend this work for publication after minor revision:

1-      Authors should present an explanation for the high net activation energy of the overall five steps (117.8 kcal/mol) compared to the non-catalyzed one step reaction (76.3 kcal/mol).

2-      Please, shorten the conclusion. It is better to be a single paragraph rather than points.

3-      Table 3 may be transferred to Supplementary data.
